# Synchronized amplification of local information transmission by peripheral retinal input

**Pablo D Jadzinsky, Stephen A Baccus***

Department of Neurobiology, Stanford University School of Medicine, Stanford, United States

**Abstract** Sensory stimuli have varying statistics influenced by both the environment and by active sensing behaviors that rapidly and globally change the sensory input. Consequently, sensory systems often adjust their neural code to the expected statistics of their sensory input to transmit novel sensory information. Here, we show that sudden peripheral motion amplifies and accelerates information transmission in salamander ganglion cells in a 50 ms time window. Underlying this gating of information is a transient increase in adaptation to contrast, enhancing sensitivity to a broader range of stimuli. Using a model and natural images, we show that this effect coincides with an expected increase in information in bipolar cells after a global image shift. Our findings reveal the dynamic allocation of energy resources to increase neural activity at times of expected high information content, a principle of adaptation that balances the competing requirements of conserving spikes and transmitting information.

## Introduction

The statistics of sensory input vary over time, due to moving objects, background motion as would arise from optic flow, and due to active sensation such as sniffing (*Shusterman et al., 2011*), whisking (*Hill et al., 2011*) or eye movements (*Tatler et al., 2006*). To achieve better performance in the current condition, many sensory systems measure the recent sensory statistics and adjust their responses to the expected sensory input. For example, adaptation in the visual system adjusts a cell's dynamic range based on the expected stimulus distribution, including the stimulus mean (*Barlow and Levick, 1969*) and variance (*Victor and Shapley, 1979*; *Smirnakis et al., 1997*; *Nagel and Doupe, 2006*). In addition, the retina adapts to spatiotemporal correlations so as to remove predictable signals and enhance the response to novel input (*Hosoya et al., 2005*).

These expectations derive from correlations in visual input, which can extend over a wide range of scales due to extended textures, motion of large objects, and from eye and body movements. For example, object motion sensitive ganglion cells receive peripheral inhibition to suppress the predictable excitatory input due to small, fixational eye movements and transmit unpredictable, novel signals from moving objects (*Olveczky et al., 2003*). In addition, inhibition from fast, large global shifts may reflect the instantaneous expectation that an eye movement is occurring, and play a role in saccadic suppression (*Roska and Werblin, 2003*; *Geffen et al., 2007*).

In addition to peripheral inhibition, it has long been known that changes in the retinal image far away from the receptive field center produce excitation (known as the 'periphery' or 'shift' effect) (*Krüger and Fischer, 1973*; *Mcilwain, 1964*) in various species, including cat, rabbit, and primate (*Watanabe and Tasaki, 1980*; *Krüger and Fischer, 1973*). The functional importance of long-range excitation, however, is unclear despite numerous studies on the spatiotemporal properties of this input. Many studies have focused on the stimulus parameters that generate excitation

*For correspondence: baccus@
stanford.edu

**Competing interests:** The authors declare that no competing interests exist.

**eLife digest** To see an object, light must travel from it and be focused onto the retina at the back of the eye. The image projected onto each retina is then processed by neurons known as ganglion cells, which transmit a processed version of the image to the brain. Each ganglion cell responds to a specific section of the retinal image, in particular to intensity changes or movements that occur within that region, known as the cell's receptive field. However, ganglion cells in the retina of many species can also become active if rapid movements occur in parts of the retinal image that are far away from the receptive field of that ganglion cell.

Jadzinsky and Baccus have now investigated how this peripheral motion affects the response of salamanders' retinal cells. The images consisted of a central object surrounded by a checkerboard pattern, the brightness of which could be varied to change the contrast of the image (higher contrast images stimulate the ganglion cells more strongly). Then, either the entire image or only the central object moved. Moving the whole image represents the pattern that would be seen if a salamander moved its eye or head to look at a new part of a scene.

Jadzinsky and Baccus found that when only the central object moved, the ganglion cells only responded to high-contrast images that strongly stimulated the cells, effectively conserving energy by only responding to strong signals. However, when the whole image moved, the cells also responded to lower-contrast images, showing that they had switched to processing the local region of the scene in more detail. These effects could be reproduced in a simple mathematical model.

The model suggests that the ganglion cells increase their information transmission at times when a large amount of new information is expected to be received: for example, immediately after the salamander has moved its eyes. The next challenges in this research are to identify the specific retinal neurons that generate this change in processing in the ganglion cells, and to further understand how sensory input influences how the nervous system allocates energy resources.

(*Barlow et al., 1977*; *Ikeda and Wright, 1972*; *Li et al., 1992*; *Passaglia et al., 2009*). However, few studies have measured how long-range excitation changes the neural code for local stimuli (*Passaglia et al., 2009*), and none has considered how image statistics might relate to such long-range excitation.

We examined how peripheral stimulation changes how a ganglion cell encodes the central part of its receptive field. Numerous studies in the salamander retina have characterized the properties of the ganglion cell receptive field center (*Smirnakis et al., 1997*; *Hosoya et al., 2005*; *Olveczky et al., 2003*; *Geffen et al., 2007*; *Kastner and Baccus, 2011*; *Werblin, 1972*), and have studied the effects of peripheral stimuli on the neural code as related to eye movements (*Olveczky et al., 2003*; *Geffen et al., 2007*; *Baccus et al., 2008*). Our experiments were performed in the isolated intact retina, and ganglion cell spiking activity was recorded using a multielectrode array.

We find that peripheral stimulation amplifies information transmission about the local stimulus in ganglion cells. Underlying this increase in information in neural responses is a more complete adaptation to the local stimulus, allowing for both low and high local contrast environments to be encoded with a similar response range. This rapid change in the neural code causes the cell to encode the intensity sequence of the stimulus and the contrast at different times relative to the global shift, thus causing peripheral motion to act as a timing signal to coordinate the encoding across a population of cells. We further show that these effects can be produced by a simple model combining local and peripheral inputs prior to a threshold and an adaptive stage. Finally, using the same model we show that the pulse of increased information that we observed when stimulating the periphery matches in timing the expected arrival of novel information generated by a global image shift as would occur during motion of a large object or an eye/head movement.

Our results show that global motion switches the neural code from one that conserves energy, encoding only strong stimuli, to one that transmits greater information and encodes both weak and strong stimuli. These findings reveal a principle of adaptation that acts to allocate energy resources in the form of neural activity to times that are expected to contain novel information.

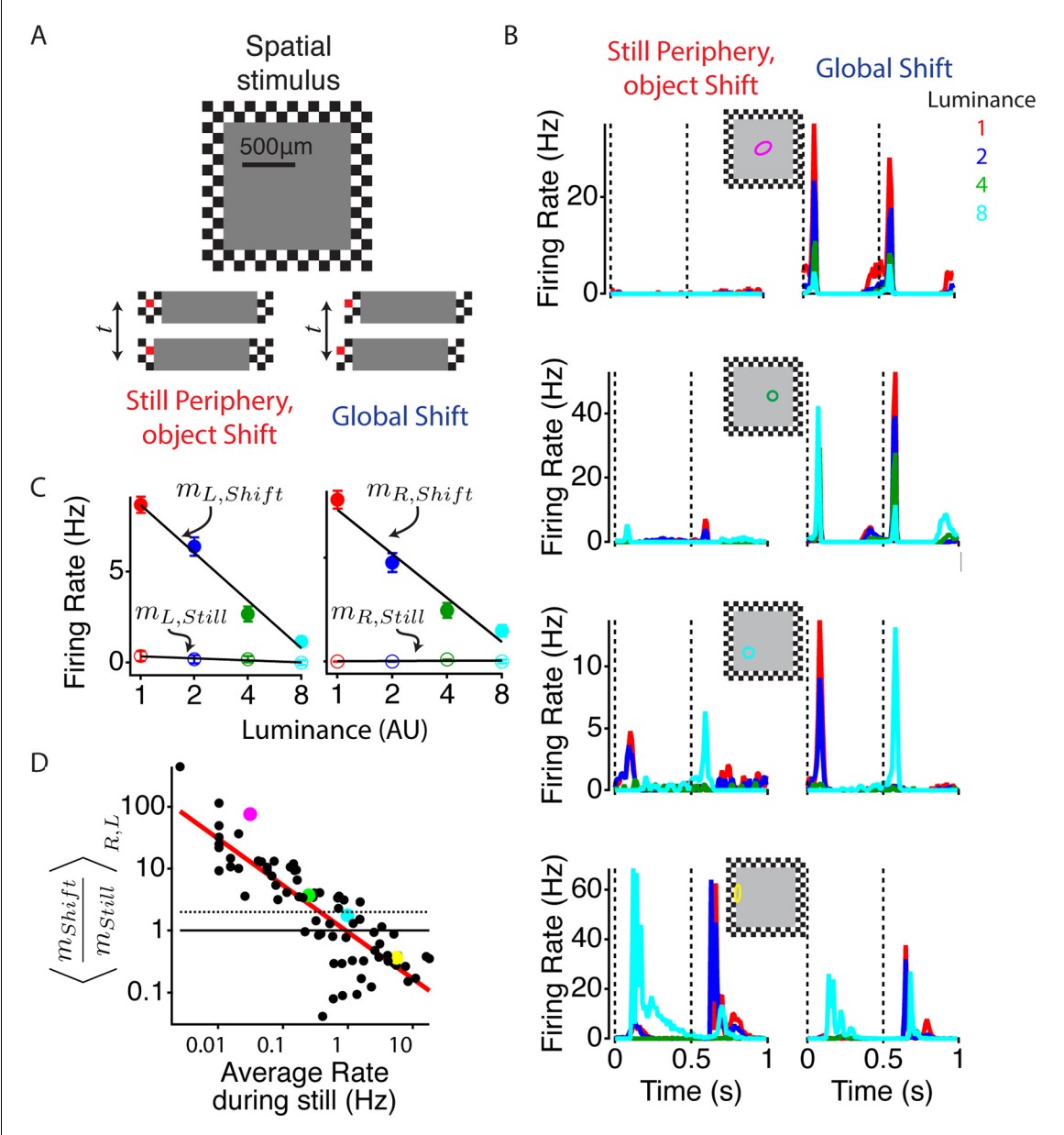

**Figure 1.** Global image shifts increase sensitivity to weak local input. (A) (Top) A diagram of the stimulus is shown. The central square representing an object shifted left or right either in the presence of a static periphery (still periphery, moving object, left in bottom panel) or in conjunction with the periphery (global shift, right in bottom panel). In both conditions, the central stimulation was the same. Shifts occurred every 0.5 s, and the luminance level in the object changed every 110 s to one of four values spaced logarithmically. Lower panel shows the central stimulus region under both peripheral conditions. One checker is colored red (not used in actual stimulus) to help the reader identify the relationship between this particular checker and the central stimulus. (B) Average firing rate response of four different cells from different preparations to four different luminance values under both peripheral conditions: object shift (left), global shift (right). Stimulus shifts to the right (0 s) and left (0.5 s) are marked with dotted lines. The classical (linear) receptive field center, computed from a white noise checkerboard stimulus is shown as a colored oval. (C) Average firing rate computed between 50 and 150 ms after the stimulus shifted to the left and right for the cell shown in (B, top panel), colors of dots show different luminance levels corresponding to the curves in (B). A linear fit (lines) to the data was used to compute the sensitivity $m$ to the luminance of the central region, computed as the slope of the firing rate vs. the log of the central luminance for left and right object shifts with periphery still (open circles, $m_{L,Still}$, $m_{R,Still}$) and for global shifts (filled circles, $m_{L,Shift}$). (D) The ratio of the luminance sensitivity $m$ during global and object shifts compared for each cell to the firing rate in the object shift condition, indicating the strength of the object shift stimulus. Axes are logarithmic. Results for $m_{Still}$ and $m_{Shift}$ were averaged over shifts to the left and right. Cells above the dotted line increased the slope of firing vs. central luminance by more than a factor of two during a global shift compared with an object shift. Colored dots correspond to the cells shown in (B).

## Results

### During global shifts, peripheral stimulation increases the response to local stimuli

To measure how peripheral motion changes the response of ganglion cells, we presented a stimulus to simulate image movement in the retina due to two different conditions: a moving object in a static scene or global motion as would be caused by movement of the eye or a large object. We chose a set of brief stimuli that could produce a wide variation in excitation – from extremely weak to very strong – depending on a cell's location, to determine the effect of peripheral excitation, and how that effect varies with the cell's central excitation. The stimulus consisted of a central square object with a constant luminance in front of a checkerboard peripheral pattern. The object covered the classical linear receptive field center plus part of the surround of most cells (*Figure 1A*, top panel). To present the same central stimulus in the presence and absence of strong peripheral stimulation, either the object alone (object shift) or the whole stimulus (global shift) was shifted abruptly by one peripheral square, 50 μm in length, ~1 degree of visual field in the salamander. We then varied the central luminance level to measure the effect of the strong peripheral stimulus in encoding the luminance of the central stimulus. When the object moved alone, many cells showed transient activity, which depended on the location of the cell relative to the object border (*Figure 1B*, left column). As expected, those cells near the center of the object showed the weakest activity because they experienced little change in light intensity, and overall, the response was insensitive to the central luminance value. In the global shift condition, however, brief strong firing events occurred during both right and left shifts for most cells including those in the center of the object (*Figure 1B*, right column). To assess the effect of the peripheral shift on the ability of a cell to distinguish different central stimuli, we computed the slope of a line fit to the firing rate as a function of the log of central intensity, and found this slope to be much greater during the global shift (*Figure 1C*). We examined which cells were most strongly affected by the periphery by comparing responses to all constant luminance objects under both peripheral conditions. In doing so, we found that the cells with the weakest response to the object condition showed the greatest enhancement of sensitivity from peripheral motion, with 39 out of 76 cells at least doubling the slope of their firing rate vs. the log of central luminance (*Figure 1D*). This increased activity during the global shift condition could not be attributed to the linear receptive field of the cell overlapping the object border, as steps to the right and left would have linear contributions of opposite signs, even if such stimuli might be within the spatial region occupied by the classical receptive field surround (*Demb et al., 2001*; *Borghuis et al., 2013*). Accordingly, the effect of the global shift was mostly insensitive to the phase of the checkers in the periphery (see Materials and methods). Thus, peripheral stimulation enhanced the sensitivity to weak central stimuli during abrupt global image motion.

### Peripheral input gates central information transmission

To analyze the dynamics of the effects of peripheral stimuli on the processing of central input, we decoupled the central and peripheral inputs by presenting a stimulus with a continuously flickering light intensity in the center, combined with brief peripheral motion. Although this stimulus differs from a global image shift, which simultaneously changes central and peripheral regions as in (*Figure 1*), the independent control of central and peripheral stimuli allowed us to assess the dynamics of how the periphery changes the encoding of a central stimulus. To create a local naturalistic input, the central object intensity flickered with a temporal power spectrum resembling natural scenes, which was inversely proportional to the frequency – called 'pink noise' (*Simoncelli and Olshausen, 2001*) (*Figure 2A*, right panel inset). The periphery was a checkered pattern that was either still, producing no temporal input, or reversed every 0.5 s, producing strong synchronized peripheral stimulation. For most recorded cells, shortly after peripheral stimulation there was an excitatory effect – indicated by an increase in firing (*Krüger and Fischer, 1973*) – followed by strong inhibition as might underlie a previously reported component of saccadic suppression (*Roska and Werblin, 2003*), and then a slower recovery to the baseline state (*Figure 2B* and *Figure 2—figure supplement 1A*). This effect also occurred with both left and right shifts of the checkers (*Figure 2B*, inset), indicating that this effect was primarily not caused by the classical (linear) receptive field surround, which would have produced opposite effects for the two background phases.

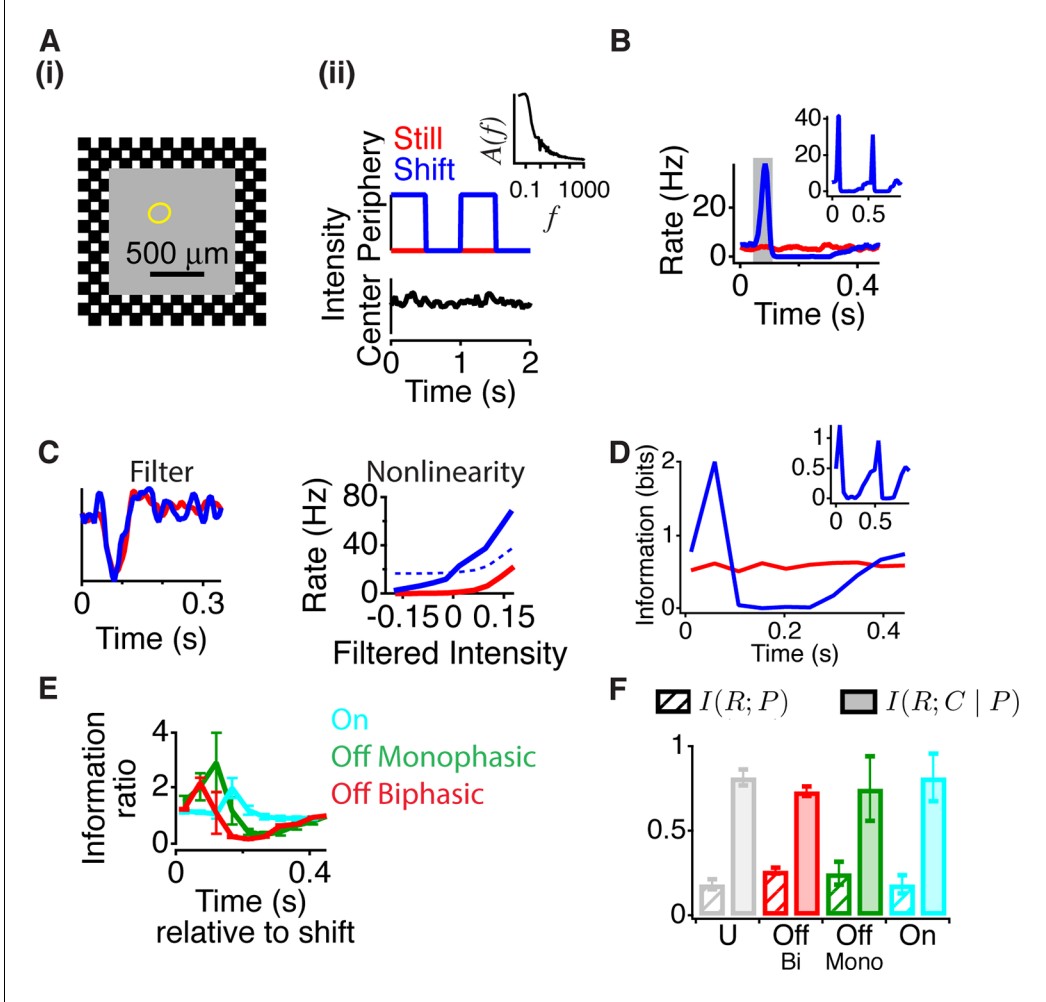

**Figure 2.** Peripheral gating of information transmission. (**A-i**) Spatial stimulus design showing central and peripheral regions. (**A-ii**) The temporal sequence in each region. The center stimulus flickered randomly with a naturalistic amplitude spectrum proportional to $1/f$ (inset). In the periphery, the stimulus either shifted (reversed in sign) every 0.5 s or was still. Most cells had linear receptive field centers fully contained in the central region (yellow oval indicates receptive field center). (**B**) Peristimulus time histogram aligned to the time of peripheral stimulation. Inset shows the two different peripheral shifts averaged separately, indicating that both excitation and inhibition occur for both peripheral phases. (**C**) Filters and nonlinearities of a linear-nonlinear model computed from the spike times and the center signal. In the Shift case, only spikes from the high firing rate window were used (gray box in B). The dashed nonlinearity is the curve that would have resulted from a vertical shift of the Still case to account for the observed increase in activity in the high firing rate window. (**D**) Mutual information between the spike count in a 50 ms time window and the central region as a function of time after the peripheral shift. Inset shows information computed separately for left and right shifts of the grating. (**E**) Average for different cell types of the normalized information in the Shift condition for three different cell types; biphasic Off (n = 95 cells), slow Off (n = 10) and slow On (n = 7). Information was normalized by the value in the last bin. (**F**) Average across cells of the information that the spike count carries about the peripheral signal $I(R; P)$ or about the central region once information about peripheral input has been removed (see Materials and methods). By the chain rule of mutual information, the two quantities add to the total amount of information the spike count conveys about the stimulus, $I(R; P, C)$.

The following figure supplement is available for figure 2:

**Figure supplement 1.** Peripheral shift increases the response to central stimuli with a natural temporal spectrum.

The presence of excitatory input, however, does not reveal how peripheral input changes the neural code for central input. For example, peripheral excitation could potentially saturate the cell, and thus mask central input. We therefore measured the sensitivity to the central region using a linear-nonlinear (LN) model (see Materials and methods), consisting of a linear temporal filter representing the average feature preferred by the cell followed by a static nonlinearity capturing the cell's average threshold and sensitivity, defined as the average slope of the nonlinearity

(*Chichilnisky, 2001*; *Baccus and Meister, 2002*). In this case, although the LN model does not capture all of the nonlinear dynamics of the receptive field center (*Gollisch and Meister, 2010*) (some of which is captured in a model below), the nonlinearity can be used as a statistical measure of sensitivity to the central stimulus (*Kastner and Baccus, 2011*; *Baccus and Meister, 2002*; *Rieke, 2001*; *Zaghloul et al., 2005*) at different times relative to the peripheral shift.

To observe changes in the neural code associated with peripheral excitation, we computed two LN models: one when the periphery was still and the second one under the shift condition using only spikes from a 50 ms gating window centered on the peak of excitation (*Figure 2C*). Even though the firing rate greatly increased during 50–100 ms after the peripheral shift, the temporal filter changed little when compared to the still condition, indicating that the cell continued to convey the same average feature about the visual stimulus during the 50 ms high firing rate interval (*Figure 2C*). We defined the sensitivity to the central stimulus as the average slope of the nonlinearity (*Kastner and Baccus, 2011*; *Baccus and Meister, 2002*) and found that the sensitivity was the greatest 50–100 ms after a peripheral shift, during the high firing rate window. This indicates that the increase in firing rate after the peripheral shift is not due to a response independent of the center stimulus, as such an effect would have shifted the nonlinearity vertically, leaving the sensitivity to the center unchanged (*Figure 2C*, dashed line). Instead, we find that an abrupt peripheral shift dynamically gates the response of a cell, enhancing the cell's sensitivity to its preferred visual feature near the receptive field center.

However, the amount of information conveyed is influenced not only by sensitivity but also by noise, and thus an increase in sensitivity does not necessarily imply an increase in information transmission. Therefore, to confirm that the increased activity and sensitivity were accompanied by an increase in transmitted information, we estimated the mutual information between the stimulus sequence in the object region and the cell's response as measured by the spike count at different times relative to the peripheral shift. This quantity is $I(R; C|p)$, the mutual information between the response, $R$, and the central intensity, $C$, given a particular peripheral stimulus $p \in P$, where $p$ is the time relative to the peripheral shift (see Materials and methods).

We found that just after a peripheral shift, information about the central stimulus sharply increased as compared to when the periphery was still, indicating that a signal from the periphery increases information transmission from the central region (*Figure 2D*, *Figure 2—figure supplement 1B and C*). After this increase, information transmission then decreased (100–300 ms after the global shift) and then recovered to the baseline state. All cell types showed a sharp increase of information during the gating window, with the peak time depending on the cell type (*Figure 2E*). The leftward shift of the nonlinearity (*Figure 2C*) increased the slope of the nonlinearity and information transmission because the threshold of the nonlinearity in the baseline condition was positioned to the right of the mean stimulus, which is the case for ganglion cells of diverse species including mammalian and primate retina (*Chichilnisky, 2001*; *Keat et al., 2001*).

In this analysis, by considering $I(R; C|p)$ we are taking the more traditional point of view that cells encode signals in the center of their receptive fields and are modified by other (peripheral) signals. However, one could argue that ganglion cells are actually encoding the periphery and being modified by the center. The total information between the response and both central and peripheral stimuli, $I(R; P, C)$ can be separated into two components by the chain rule for mutual information (*Cover and Thomas, 1991*) (see Materials and methods).

$$\begin{aligned} I(R; P, C) &= I(R; P) + I(R; C|P) \\ &= I(R; P) + \langle I(R; C|p) \rangle_{p \varepsilon P} \end{aligned} \tag{1}$$

However, on average, for all cell types studied, the information that the cell carried about the peripheral stimulus $I(R; P)$ was substantially smaller than the information the cell carried about the center given the peripheral stimulus ($I(R; C|P)$), averaging 27% of $I(R; C|P)$ (*Figure 2F*). These results support the view that peripheral input gates information transmission from the central region.

Although it may seem puzzling that little information is conveyed about the periphery even though there is a large timed increase in the average firing rate, this is because the large peak at ~100 ms represents only the average response to the stimulus. On any given trial, the decision to fire is primarily controlled by the central input and in many cases, no spikes occurred when the periphery moved. Nonetheless, the brain could use a population of cells to identify the gating

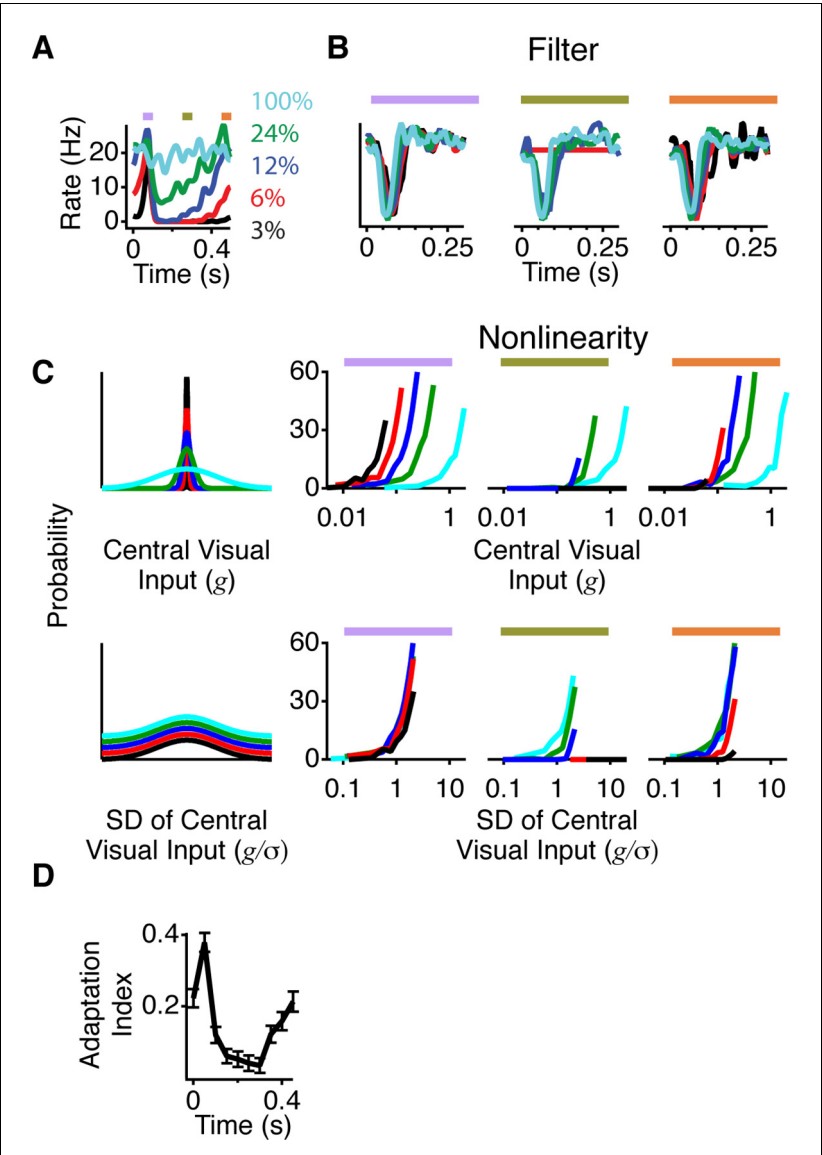

**Figure 3.** A more adapted representation underlies an increase in information transmission. (**A**) The spatial stimulus was the same as in *Figure 2*. The time course of the central stimulus was a Gaussian white noise stimulus with one of four different contrasts or 100% binary contrast, consisting of black and white intensity values. PSTHs are shown for the different conditions. (**B**) Filters computed using only spikes from 50 ms time windows, corresponding to color boxes in (**A**). Purple, gating window; Olive green, suppression window; Orange, recovery window. (**C**) Input distributions (left) and nonlinearities in the same three 50 ms time windows as in (**B**). Upper curves are all in units of the linear prediction; lower curves show the same data but in units of standard deviation of the linear prediction. The abscissa is displayed on a logarithmic scale, such that normalization by the standard deviation produces a lateral shift. (**D**) Average adaptation index across cells that exhibited peripheral excitation (see Materials and methods, n = 400).

The following figure supplement is available for figure 3:

**Figure supplement 1.** Periphery induced changes in adaptation.

window as a time when activity increases synchronously throughout the retina. Our analysis using the number of spikes in a time bin neglects more complex encoding due to latency (*Gollisch and Meister, 2008*) and firing patterns in the population (*Schnitzer and Meister, 2003*). Yet, this analysis

suggests that the brain could extract this increased information simply by counting spikes, without the need for a more complex decoding scheme across time bins or using the population.

## Information transmission increases to half the maximal value

We then compared the amount of information transmission with the theoretical maximum given the cell's stochastic firing properties. We assessed the theoretical maximum by computing the sigmoidal nonlinearity that maximized information about the stimulus, given a cell's maximum firing rate and measured spiking noise as defined by its spike-count distribution for a given average response, $P(r|\langle r \rangle)$. The average firing rate however was not constrained at a particular value, in contrast to previous studies, meaning that a leftward shift of the nonlinearity with the same maximum would generate a higher average firing rate (*Pitkow and Meister, 2012*) (see Materials and methods). After a shift, the information conveyed was 0.47 ± 0.04, (n = 18 cells) of the maximal value, compared with 0.24 ± 0.05 before the shift. Thus, after the shift, the neural code used more of the capacity of the cell given its noise properties and the spike count code. However, this came at the increased cost of energy in terms of a higher firing rate (6.9 ± 1.7 Hz after the shift, and 1.9 ± 0.3 before the shift). Previous results indicate that the high threshold of ganglion cells allows them to conserve spikes at the expense of maximal information transmission (*Pitkow and Meister, 2012*). Our analysis indicates that after a peripheral shift, the neural code shifts away from energy conservation and towards high-throughput information transmission.

## Peripheral stimulation gates a change in adaptation

To examine which properties of the neural code changed between the shift and still conditions, we presented a Gaussian white noise sequence with a fixed mean and different contrasts, defined as the standard deviation divided by the mean. This allowed us to compute and compare separate LN models for each contrast. We observed that both excitation and inhibition from a peripheral shift depended on the central contrast, with a much stronger increase in firing observed at low contrast (*Figure 3A*). This result is consistent with the observation that cells with the weakest response to a moving square had the strongest effect from peripheral stimuli (*Figure 1*). We then fitted LN models as stated above at different contrasts during 50 ms time windows corresponding to gating, suppression, and recovery (*Figure 3B–C*). As the contrast in the central region decreased, the filter in some cells became slower and more monophasic as previously reported (*Baccus and Meister, 2002*). However, although at low contrast, the gating window's firing rate exceeded the recovery window's firing rate by more than 20-fold, the filters were markedly similar. Thus, peripheral stimulation affected the sensitivity, but had a minimal effect on the average local features preferred by the ganglion cells.

Changes in sensitivity are also known to be caused by adaptation to the local contrast. We therefore tested how peripheral stimulation influenced adaptation to the central contrast by measuring changes in adaptation as a function of time since the peripheral shift. For an ideally adapting cell, the sensitivity would scale in inverse proportion to the contrast (*Figure 3—figure supplement 1A–B*). Previous results, however, have shown that ganglion cells adapt less than this ideal amount, in particular at low contrasts (*Ozuysal, 2012*). We found that in the gating window, responses to different contrasts were much more similar to each other, indicating a greater level of adaptation to the central contrast (*Figure 3C*). This effect arose because at low contrast, the slope of the nonlinearities changed more than at high contrast, similar to the stronger effects of gating seen with cells that responded more weakly to a shifting object (*Figure 1D*). At 3% contrast, the slope changed by 5.5 ± 1.1 Hz per contrast unit (one s.d. of the nonlinearity input was 0.03 contrast units, or 3% ) and at 24% contrast the slope changed by 0.20 ± 0.06 Hz per contrast unit (one s.d. was 0.24 contrast units, or 24% ) (*Figure 3—figure supplement 1C*). As a result of these effects at different contrasts, during the gating window nonlinearities reached a more similar height across contrasts. Accordingly, when normalized by the stimulus standard deviation at each contrast, nonlinearities also had a more similar shape across contrasts in the gating window. We computed an index of adaptation that takes the value of 1 for an ideally adapting cell and 0 for a non-adapting cell (see Materials and methods and *Figure 3—figure supplement 1A–B*), and found that most cells increased their adaptation to contrast during the gating window, such that all contrasts were represented with more similar responses than during the recovery window (*Figure 3D*). This indicates that an increase in adaptation underlies

the increase in information during the gating window, such that near the receptive field center, both weak and strong signals are conveyed.

## Intensity sequence and contrast are encoded serially after a peripheral shift

Cells exhibiting contrast adaptation – by changing their sensitivity when the contrast changes – will increase information about fluctuations in intensity, but potentially lose information about the contrast itself (*Fairhall et al., 2001*). Because many cells showed increased adaptation after a peripheral shift, we tested whether the cells encoded different properties of the stimulus – the sequence of light intensities and the contrast – at different times relative to a peripheral shift. We designed a flickering stimulus that had a relatively small number of conditions to facilitate the estimation of information about the stimulus sequence and/or the contrast. The center followed a binary white noise *M*-sequence, at four possible contrasts $\sigma$, where the instantaneous intensity value was $\mu + \sigma \cdot m$ and $\mu$ is the mean intensity, fixed throughout the experiment and $m = \pm 1$ are the instantaneous values of the M - sequence (see Materials and methods). All combinations of binary sequences (up to four frames, lasting 400 ms, $m^{(4)} \in M^{(4)}$, contrasts and times relative to peripheral excitation were presented an equal number of times (see Materials and methods and *Figure 4A*). *Figure 4B* shows raster plots for one-cell ordered according to contrast and mixing the responses to all M - sequences. We estimated the mutual information between the response and two stimulus parameters at different times relative to the peripheral shift – the light intensity sequence in the previous four frames $(M^{(4)})$, $I\left(R; M^{(4)}|p\right)$, and the center's contrast $(\sum)$, $I(R; \sum |p)$, where $\sigma \in \sum$ (*Figure 4C*). We found that the responses coming from the same cell at different times carry different types of information. When computing $I\left(R; M^{(4)}|p\right)$, the analysis was conducted as if the brain was decoding the stimulus sequence without knowing the contrast. The results were similar, however, if we considered that the brain might decode the contrast, and use this knowledge to better decode the stimulus sequence (eq (10) and *Figure 4—figure supplement 1*). Whereas a static neural code would typically face the choice between adaptation and preserving the adapting statistic, a dynamic neural code avoids this tradeoff by rapidly switching between complementary representations of the same stimulus.

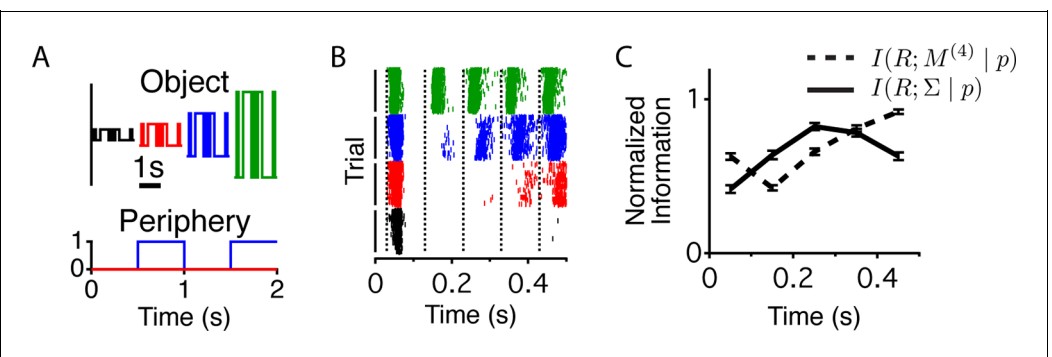

**Figure 4.** Different stimulus properties are conveyed with different dynamics. (**A**) Experimental design for the measurement of sequence and contrast information. The center object follows a binary *M*-sequence at four different contrasts. Each position in the sequence and contrast combination occurs at all possible times relative to a peripheral shift. (**B**) Raster plots for an example cell aligned to the time of the peripheral shift and ordered according to contrast. Many different sequences are shown for each contrast value. Luminance values in the center change every 100 ms, generating temporally discrete responses. Vertical lines show the times used to extract responses for the information calculation. (**C**) Average across cells (n = 94) of the normalized information conveyed about the contrast (four different levels, solid line) or the four frame stimulus sequence *M*<sup>(4)</sup> (dashed line) as a function of time since the shift.

The following figure supplement is available for figure 4:

**Figure supplement 1.** Different components of the stimulus are conveyed with different dynamics.

## A simple model produces gating and changes in adaptation

To identify the minimum components required to produce the dynamic neural code we observed, we began with the known structure of excitatory input to a ganglion cell consisting of a central pathway comprising a linear filter, a static nonlinearity and an adaptive stage. This central pathway model was analogous to an accurate model of contrast adaptation (*Ozuysal, 2012*), where the rectified nonlinearity and adaptation likely occur at the bipolar cell synaptic terminal, although here we used a simplified adaptive stage. Nonlinear peripheral input is delivered only in the inner retina, as horizontal cells do not respond to fine gratings (*Baccus et al., 2008*). Therefore, the only remaining choice in the model structure was the level at which to combine the peripheral input. Because a peripheral shift caused the overall nonlinearity of the LN model to shift laterally, rather than vertically with respect to the central pathway's linear input (*Figures 2* and *3*), the peripheral pathway delivers input prior to the nonlinearity, corresponding to input to the bipolar cell terminal (*Figure 5A*). The peripheral pathway can be represented by small rectified subunits that cause the response to be insensitive to the peripheral pattern (*Victor, 1979*; *Olveczky et al., 2003*). Rather than explicitly simulate spatiotemporal dynamics of the periphery, we modeled the net effect of the abrupt peripheral stimulus via a timed biphasic signal. The model effectively replicated the data for Gaussian stimuli at different contrasts (*Figure 5C, D*), as well as for constant luminance stimuli (*Figure 5—figure supplement 1*).

A key aspect of the model is the order in which the signals are combined. Because the peripheral input is delivered prior to the threshold and adaptive stage, it is summed with the unadapted measure of the central input. This causes the peripheral input to have a larger effect on weak central stimuli than on strong ones as experimentally observed (*Figures 1* and *3*). Furthermore, weak stimuli only cross threshold when peripheral input is applied (*Figure 5B*), allowing the adaptive stage to encode and adapt to signals of all strengths, including those from a central stimulus with a constant intensity (*Figure 5—figure supplement 1B*). Thus, when peripheral excitation is present (*Figure 5B*, *Figure 5—figure supplement 1A*), the model applies a lower threshold relative to the central input, conveying more information about the full range of contrast environments, but less information about the contrast level. When peripheral input is absent or inhibitory, the model applies a high threshold with respect to the central input (*Figure 5B* and *Figure 5—figure supplement 1A*), conveying primarily higher contrasts and encoding information about the contrast level.

## Peripheral input is independent of central contrast

An important conclusion we derived from the model is that a single amplitude of peripheral input (equivalent to a fixed threshold shift with respect to the central input) replicates the results at all central contrasts. Thus even though the effects of peripheral input differ with the central contrast, this is because of the differing downstream effects of adaptation; the peripheral signal itself delivered prior to the threshold does not depend on the central contrast.

## Peripheral excitation and inhibition act across a similar spatial scale

We then measured the scale of excitation and compared it to the scale of transient peripheral inhibition, which is thought to play a role in suppressing the effects of eye movements (*Olveczky et al., 2003*; *Roska and Werblin, 2003*). To measure the distance over which lateral excitation acts, we designed a stimulus to measure peripheral gating of sensitivity to a central object as a function of distance from the peripheral stimulus. The stimulus had three different components. First, the periphery was composed of 50 µm checkers that covered the whole screen. Second, on top of the periphery, a mean intensity gray mask covered the peripheral checkers over the central region; the size of the mask, *L,* was varied. Third, the central object consisting of a pink noise flickering sequence was then added on top of the central gray region and was presented as a fenestrated checkerboard pattern of fixed size (*Figure 6A–B*). By decreasing the mask size, more peripheral checkers were present, and the distance between peripheral checkers to any measured cell was decreased. At the smallest mask sizes, peripheral checkers were intercalated with the central region object, and occupied all space not covered by the central object (*Figure 6B*, bottom). The excitatory influence of gating acted at distances of up to 1 mm (*Figure 6C*). This further confirms that the effect was distinct from the linear receptive field, which would not be activated by a fine checkerboard at

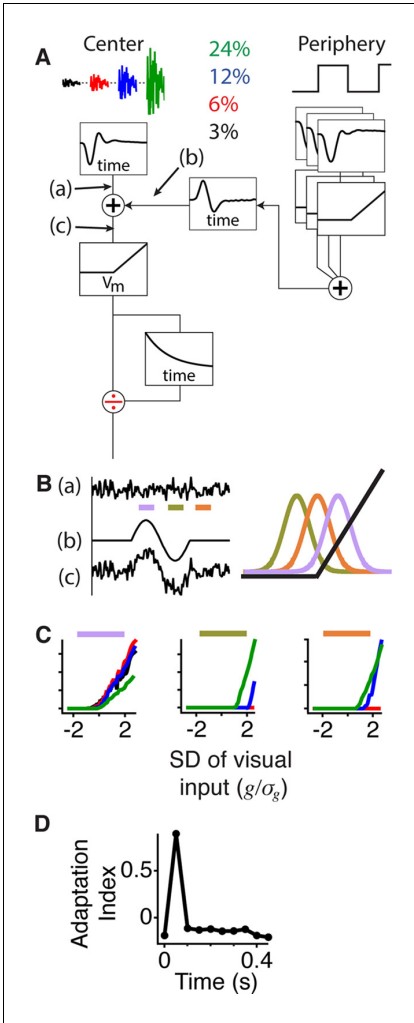

**Figure 5.** Gating of information through a shift in an internal threshold. (**A**) Model of a cell where two pathways are combined prior to a threshold and an adaptive block, implemented here as a feed forward divisive effect with a memory. Peripheral pathway is composed of many nonlinear subunits making the pathway insensitive to the stimulus spatial pattern and delivers biphasic input to the central pathway (first positive, then negative). The stimulus is Gaussian white noise at 3–24% contrast matching the experiment (and colors) in *Figure 3*. The central pathway is composed of a linear temporal filter, because stimulus is only a function of time. (**B**) Signals arising at points (**a**), (**b**) and (**c**) whose locations in the model are marked in panel (**A**). When the peripheral input is positive (gating window, purple bar) or negative (suppression window, olive green bar) the central input is shifted to higher or lower values with respect to the baseline state (recovery window, orange bar) and fixed threshold. Right, the Gaussian distribution of the filtered stimulus occurring at point (**c**) in (**A**) compared to the threshold nonlinearity occurring after point (**c**) in (**A**) during the gating (purple), suppression (olive green) and recovery (orange) windows. The peripheral input effectively shifts the Gaussian mean with respect to the fixed threshold. (**C**) Model responses to the same Gaussian stimulus used in *Figure 3* at the times of the corresponding color bars in (**B**). (**D**) Adaptation index for the model's output as a function of time after the shift.

The following figure supplement is available for figure 5:

**Figure supplement 1.** Model responses to peripheral gating for steady central stimuli.

this distance. The observed changes in sensitivity indicate that excitatory and inhibitory influences acted over a similar scale, equivalent to a radius of ~20 deg. of visual angle.

## Discussion

Our results show that global image shifts generate a signal that increases information transfer about a cell's preferred visual feature from local regions near the receptive field center. In the gating time window, a change in the neural code is explained by a simple additive signal prior to a threshold followed by an adaptive stage. As a result, a broader range of signals is transmitted, with a stronger effect on weak, low-contrast stimuli commonly found in natural scenes (*Tadmor and Tolhurst, 2000*). This gating of information transmission, caused by the peripheral signal, results in more complete adaptation to the image contrast as compared to other times. We explain the functional importance of this effect as the resolution of two competing needs: conservation of spikes and the rapid encoding of information when it is expected that information in the center will suddenly increase.

### A model compares the timing of gating with the expected increase in information after a global shift

Our results indicate that transient peripheral input increases information transmission about the visual feature in the receptive field center. However, for natural images and an abrupt global shift in the image as from motion of a large object, the eye or head, the transient effect of gating would be combined with transient changes in image statistics created by the image shift. We therefore compared the timing of our gating results (*Figure 2*) to the dynamics of the expected information transmission for natural scenes during an abrupt shift in the image. The statistics over the receptive field center depend both on the image and the sequence of eye movements, and have strong correlations over time. Consequently, attaining representative statistics of the distributions of both light intensity and responses over multiple time points requires many trials. Owing to the difficulty in sampling the high dimensional distribution of the stimulus and responses over multiple time points (see Materials and methods), such an experiment would be prohibitively long. We therefore addressed this question using a spatiotemporal version of our model of gating with simulated eye movements and a large number of natural images.

To estimate the timing of information transmission under natural images, we combined natural images with simulated global shifts of the image (*Figure 7A-i*). The spatiotemporal model used for fast Off-type ganglion cells had the same structure as the reduced model (*Figure 5*), but we replaced the initial linear filter with the spatiotemporal receptive field measured from a fast Off-type bipolar cell (*Baccus et al., 2008*). Because peripheral gating is delivered prior to the threshold in the model, it is likely that it represents an input presynaptic to the bipolar cell terminal. Furthermore, because strong adaptation to contrast is thought to arise in the presynaptic terminal, the model is effectively of the synaptic release from bipolar cells, although the actual density of bipolar cells was not modeled. Thus, although further nonlinearities exist in the inner retina that would make the responses of ganglion cells to natural scenes more complex, we expect this model will be informative as to the relative timing of bipolar cell release and peripheral gating. Fast Off-type bipolar cells are roughly linear at a constant mean intensity for a stimulus that flickers (*Figure 7—figure supplement 1*) or jitters as in fixational eye movements, and previous models indicate that these cells may convey the primary input to fast Off-type ganglion cells (*Baccus et al., 2008*). Because the model was used to estimate the information transmitted after a global image shift, we measured the noise in bipolar cells at different contrasts. The signal-to-noise ratio (SNR) increased with contrast, which was incorporated into the model ( *Figure 7A-iii*, see Materials and methods).

This model does not capture all nonlinearities of the bipolar cell response, including luminance adaptation, a slightly saturating nonlinearity for high contrast stimuli, and weak contrast adaptation. Furthermore, because we did not include luminance adaptation, the model effectively assumes that during the fixation period, adaptation has reached a steady state, and that the global shift is too brief to cause substantial luminance adaptation. The main goal of the model, however, was to gain insight into the dynamics of information transmission under sudden image shifts.

The input to the model consisted of a set of natural images combined with fixational drift eye movements interrupted by a sudden image shift of 6 degrees (*Figure 7A*), sufficient to exceed most local image correlations (*Figure 7—figure supplement 1A*). Because natural images have strong correlations, and because bipolar cell receptive fields are biphasic both in space and time, fixational drift created a relatively small change in the filtered stimulus (*Figure 7A-ii*). However, for a brief

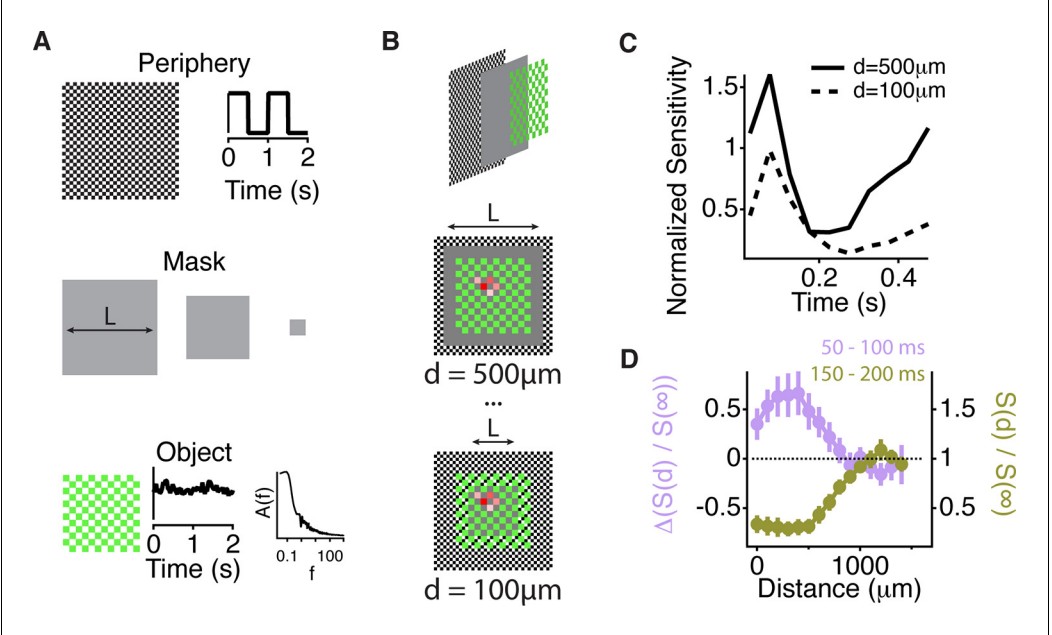

**Figure 6.** Similar spatial scale for peripheral excitation and inhibition. (**A**) Experimental design for measuring the spatial scale of peripheral excitation and inhibition (see Materials and methods). (Top) The periphery was a checkerboard pattern with squares of 50 μm covering all the screen that reversed in intensity at 1 Hz. (Middle) A gray mask with no temporal component and variable size, *L*, was drawn on top of the checkers. (Bottom) The object was a checkered pattern with a square size of 100 μm (shown in green for illustration). Object squares in the center flickered with a pink noise distribution, with an equivalent contrast of 10% , changing every 30 ms. (**B**) Top, schematic of how the different components of the stimulus were layered. Middle and Bottom, a sample cell's spatial receptive field for the object stimulus is shown in red with the color representing sensitivity to that particular square of the object for two different sizes of the intermediate mask. With this design, the object does not change across the different conditions and any change in the sensitivity to the object is due to the distance of the peripheral checkers. (**C**) Average over cells (n = 66) of the normalized sensitivity to the object stimulus, which was computed as the average slope $S(d, t)$ of the nonlinearity of a linear-nonlinear model normalized by the average slope of the nonlinearity when the background was at infinity, $S(\infty, t)$ as a function of time bin $t$, relative to the background shift for two different mask sizes. (**D**) Average of the normalized sensitivity as in panel (**C**) as a function of distance between the cell and the mask during the gating window (50–100 ms after the shift) and an inhibitory window (150–200 ms after the shift). Each point in a line corresponds to the minimum distance between the cell's linear receptive field and the background checkers for a particular background condition mask L. For the gating window, the baseline of sensitivity at 0–50 ms, which is too soon after the shift to be affected by it, was subtracted for each distance d. This subtraction was not done for the recovery window because at distances less than 500 μm, residual inhibition creates a saturating decrease in sensitivity, causing many cells to have zero slope at this time. See *Figure 6—Figure supplement 1* for sensitivity before the subtraction of this baseline.

The following figure supplement is available for figure 6:

**Figure supplement 1.** Similar spatial scale for peripheral excitation and inhibition.

window of time after the shift, the light intensity seen by the bipolar cell was less correlated with its previous values (*Figure 7B, D and F*). During this window, a wider range of possible filtered stimuli occurred, but still with most values remaining close to zero and under the fixed threshold (*Figure 7A-ii, B*). Based on the noise measurements in bipolar cells, because of the higher variance after the shift, the SNR of the bipolar cell membrane potential would be expected to increase. Note that these simulations do not include the effects of moving objects in the environment, which would cause the stimulus to more closely resemble experiments in *Figures 2–4*, where central and peripheral inputs are uncorrelated.

We first estimated the mutual information between the set of linearly filtered stimuli, $G = \{g(t)\}$, and the model's response, $R = \{r(t)\}$, as a function of time from the shift (*Figure 7C*, dashed line). Because of strong correlations in the stimulus (discussed further below), sequential measurements of the intensity in a new environment are largely redundant, and thus may not convey added information. To account for this effect, we estimated the novel information learned about the stimulus from a response given that the system has access to the previous response. This is the conditional mutual information $I(G_t; R_t | R_{t-\Delta}, p)$ between the set of linear predictions $G_t$ and the responses $R_t$ at the

same time $t$ given the response at the previous time point $R_{t-\Delta}$, computed for a given delay $p$ from the shift (see Materials and methods). As expected, because during fixational eye movements responses are highly correlated in time, each additional sample conveyed little novel information. However, after the sudden shift moved the receptive field to a new location, the conditional mutual information abruptly increased (*Figure 7C*, solid line). Unlike the mutual information at different times relative to the shift, $I(G; R|p)$, the conditional mutual information captures the intuition that most of the information about the new environment should occur in a short amount of time (*Figure 7—figure supplement 2*). Most importantly, the conditional mutual information showed faster dynamics (*Figure 7C*, continuous black line), with new information arriving faster than the peak of the linear prediction (*Figure 7C*, green line).

We then compared the timing of the increase in information from gating (*Figures 2–3*) with the timing of the expected increase in the conditional information following the shift (*Figure 7D*). We found that these greatly overlapped, meaning that the active increase in information produced by gating is timed to match the expected increase in novel information generated by the shift. Given the statistics of natural scenes and the measured noise in bipolar cells, our model indicates that gating represents a mechanism that increases information transmission at the expected peak of novel information after a global shift in the image.

## A principle of adaptation that dynamically balances information transmission and energy conservation

Adaptation is typically considered to be a process that optimizes information transmission given the current environment, and previous studies have focused on which threshold response curve would maximize information in the current environment (*Laughlin, 1981*; *Brenner et al., 2000*). However, it is clear that information transmission is not the only objective, as the threshold of retinal ganglion cells is much higher than predicted by this ideal. Consequently, it has been proposed as an alternative factor that ganglion cells conserve spikes at the expense of maximal information transmission (*Pitkow and Meister, 2012*). We propose that neither view of a neural code optimized for a single current environment – either for maximal information transmission or for energy efficiency – is fully representative of natural vision. Our findings indicate that a peripheral shift causes a switch from a code that conserves energy to one of increased information transmission, with higher information transmission occurring at the expected time of higher signal-to-noise ratio and higher information content. These results suggest a general principle of neural coding – the dynamic allocation of neural activity to times most likely to contain novel information. This principle of adaptation acts to allocate resources across environments, and in fact is analogous to known principles of communication theory that act to enhance dynamic information transmission under an energy constraint.

## Allocation of power in communications theory

An energy-efficient communications channel that carries signals over a range of frequencies should allocate power such that signal power plus noise power is a constant, except for frequencies where noise exceeds a value set by the available power (*Shannon, 1998*; *Warland and Rieke, 1999*). This concept of efficient power allocation is known as 'water-filling', as suggested by the notion of pouring power allocated to signal transmission into a basin whose depth varies according to noise but whose surface level is constant.

It is less well appreciated in neuroscience that the same water-filling principle applies to efficiently allocate power over time when the noise level dynamically varies as can occur during wireless transmission (*Goldsmith and Varaiya, 1997*). In this case, greater power should be allocated to times of higher SNR. Because a higher variance signal has a higher SNR in the bipolar cell (*Figure 7A-iii* [*Ozuysal, 2012*]), this principle agrees with our observation that additional spiking is allocated to times of expected high variance of the filtered stimulus (*Figure 7C–D*). However, because both signal and noise are changing dynamically, further studies are needed to compare dynamic changes in information transmission to an estimated optimal allocation of power over time given the changing stimuli and noise.

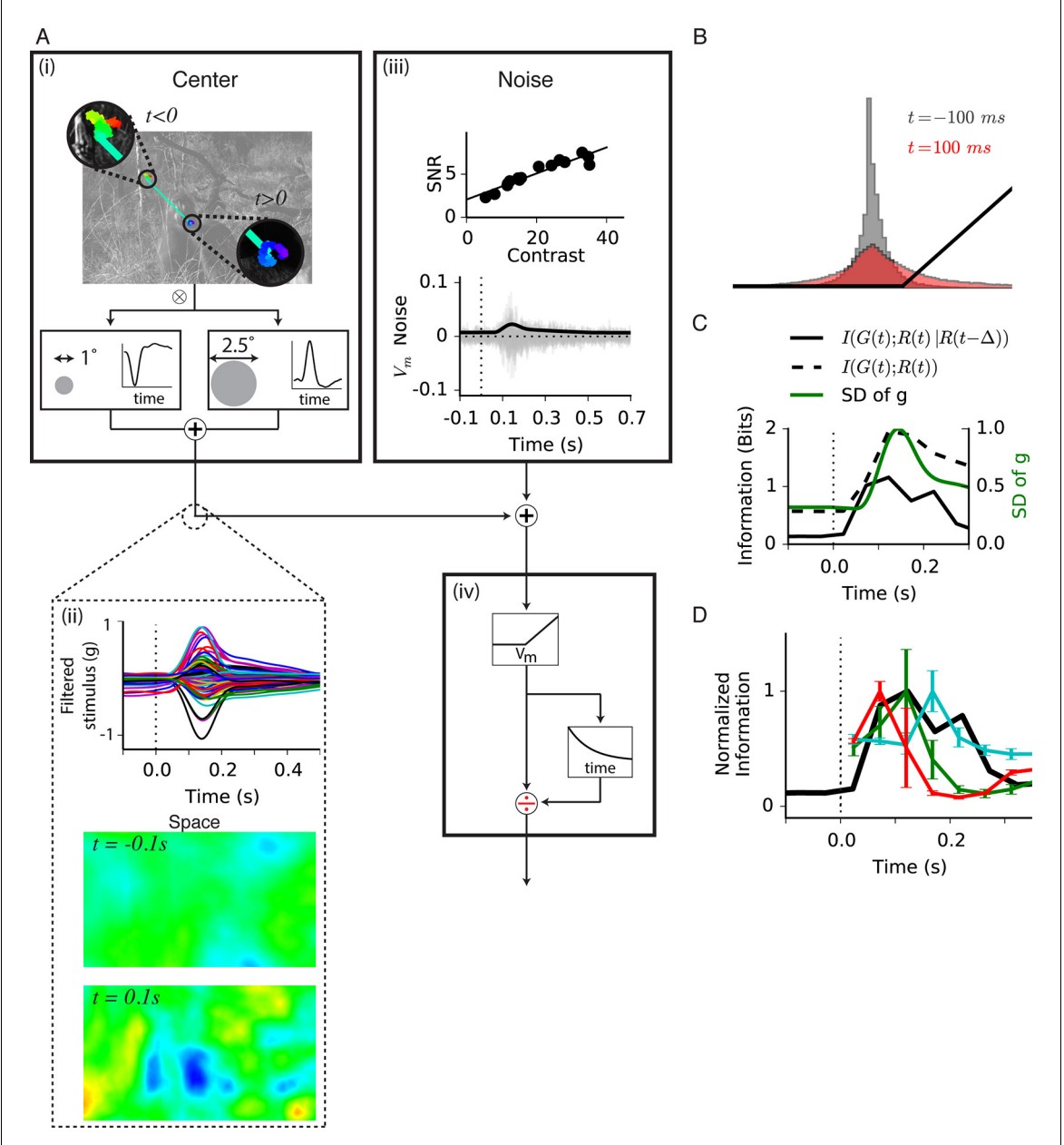

**Figure 7.** Model of central information for natural scenes with eye movements. (**A**) Spatiotemporal model used in the simulation. (**A-i**) An eye movement trajectory overlaid on a natural image, consisting of fixational drift, and a sudden eye movement (green line) that takes the center of each cell from one image location (insets) to another. The image series is convolved (⊗) with a separable spatiotemporal filter previously measured from a fast Off-type bipolar cell (**Baccus et al., 2008**), yielding a linear prediction for a bipolar cell at each spatial location. (**A-ii**) (Top) The linear prediction for 100 model bipolar cells over different images as a function of time. A sudden eye movement occurs at 0 s (dotted line). Vertical scale is in arbitrary units. (Middle and Bottom) The linear prediction is shown for a population of bipolar cells (one at each spatial location) at 100 ms before and after the sudden image shift responding to the image and eye movement trajectory shown in (**A-i**). Color scale is the same in both images. (**A-iii**) Noise model. (Top) signal to noise ratio measured experimentally in a bipolar cell from repeats of a spatially uniform Gaussian white-noise stimulus under different stimulus contrasts (**Figure 7—figure supplement 1**) (**Ozuysal, 2012**). (Bottom) Noise generated with this model, shown in the same arbitrary units as in (**A-ii**) Top. Black line is the standard deviation of the noise at each point in time. (**A-iv**) (Top) After the linear central input is summed with the noise, the result is passed through a rectifying nonlinearity and then through a feedforward divisive operation representing a simplified version of adaptation, as in the model in **Figure 5**. (**B**) Distributions of linear prediction values over many images at different times, compared with the rectifying nonlinearity (black line) from (**A-iv**). Distributions before t = 0 s and after $350\ ms$ are identical. (**C**) Dynamics of information transmission after a sudden eye movement. The Shannon mutual information $I(G_t; R_t|p)$ between the linearly filtered stimulus, $g(t)$, and the model output, $r(t)$ (black dashed line) at a given delay from the shift, $p$, and the conditional mutual information $I(G_t; R_t|R_{t-\Delta}, p)$ between the same quantities when conditioning on the response

*Figure 7 continued on next page*

*Figure 7 continued*

at a previous time $r(t - \Delta)$ (black solid line). Both stimuli and responses were averages over bins of 50 ms. Also shown is the standard deviation of the linear prediction from (**A-iii**) (green line). (**D**) Comparison of the expected conditional mutual information from the model at each time after an image shift (black line) with the time course of information transmission measured during experiments for several cell types (reproduced from *Figure 2E*).

The following figure supplements are available for figure 7:

**Figure supplement 1.** Noise measurements in bipolar cells.

**Figure supplement 2.** Correlations and information in a spatiotemporal model of gating.

## Potential role of peripheral gating during eye movements

Our results indicated that gating occurs at the time when the expected statistics of the central input will change. This expectation may arise from the sudden movement of large objects, and we expect that peripheral gating will be important during natural saccadic eye movements. Previous results have strongly implicated transient nonlinear peripheral inhibition in suppressing the effects of fixational eye movements on object motion sensitive ganglion cells (*Olveczky et al., 2003*), and the similar spatial scale of peripheral nonlinear gating and nonlinear inhibition (*Figure 6*) is consistent with a role of gating in eye movements.

Although salamanders and other amphibians have differences in their eye movements in that they make head saccades to target prey (*Manteuffel and Roth, 1993*), they have similar fixational drift (*Manteuffel et al., 1977*) and optokinetic head movements (*Manteuffel, 1984*) to mammals. Accordingly, the property of object motion sensitivity related to fixational eye movements is common to both salamanders and mammals. Similarly, the basic phenomenon of peripheral retinal excitation has been observed in mammals, and we expect that effects on neural coding we observe here will be similar. The duration of the ~1 degree global image shifts we have used (*Figure 1*) is one stimulus frame (~30 ms), similar to the duration of a ~1 degree saccade in a number of species, for rabbit, cat and monkey, 20–50 ms (*Collewijn and Zuidam, 1977*; *Evinger et al., 1981*; *Fuchs and Johns, 1967*); humans, ~20 ms (*Baloh et al., 1975*); and fish, ~70 ms (Easter, 1975). Although larger saccadic eye movements are longer in duration, the key property we find is that the timing of the linear filter in the receptive field center is coincident with temporal filtering from the peripheral input. Thus, even if the global shift is more smooth as in the case of a larger saccade or an amphibian head saccade, we expect that the excitation from both center and periphery will still coincide.

## Synchronization signals and dynamic allocation of sensitivity

Timing signals that indicate a changing stimulus have been observed in other systems that use active sensation, including sniffing in olfaction and whisking in the vibrissae system (*Shusterman et al., 2011*; *Hill et al., 2011*). In these cases, an efferent copy of a motor command can provide the timing signal. But because the retina lacks such an efferent copy, a signal that the stimulus is changing must be computed from the sensory input. Inhibitory amacrine cells are known to deliver signals laterally across long distances, and could increase the firing rate through synaptic disinhibition (*Barlow et al., 1977*). We note that a similar organization is found in the hippocampus, where a common signal is generated by oscillations in inhibitory neurons (*Buzsáki, 2002*). On the principle that the threshold should be lowered when greater information is expected, synchronous oscillations between brain regions may perform a similar function of allocating sensitivity to time intervals of greater information content.

## Tradeoffs in the neural code

The neural code embodies a choice between tradeoffs. A high threshold may be efficient in terms of energy, at the expense of the amount of information (*Pitkow and Meister, 2012*). A biphasic filter and a threshold may emphasize novelty in natural scenes (*Srinivasan et al., 1982*), but certain stimuli such as a constant luminance will be rejected. An adaptive system may improve information transmission across an entire set of stimuli, but the particular statistic that triggers adaptation may be lost. It is commonly assumed that a cell makes a single choice among these alternatives, whatever

the benefits and consequences. Our results, however, show that cells can sequentially switch between complementary representations to capture the benefits of both.

## Materials and methods

### Electrophysiology
Larval tiger salamander retinal ganglion cells were recorded using an array of 60 electrodes (Multi-channel Systems) as described (*Kastner and Baccus, 2011*). Intracellular recordings from bipolar cells were performed using sharp microelectrodes as previously described (*Ozuysal, 2012*)

### Visual Stimulus
A video monitor projected stimuli at 60 Hz, and values of intensity changed at 30 Hz. The monitor was calibrated using a photodiode to ensure the linearity of the display. Stimuli had a constant mean intensity of ~10 mW/m$^2$. Contrast was defined as the standard deviation divided by the mean of the intensity values, unless otherwise noted.

### Moving objects versus global shifts
To measure the difference between object and global shifts (*Figure 1*), the stimulus consisted of a square object 1200 μm on a side and a constant luminance of one of four logarithmically spaced values, and was presented in front of a black and white background checkerboard (50-μm squares). Either the object alone or the entire image was suddenly displaced 50 μm left and right at 1 Hz. The experiment was repeated with both phases of the background checkerboard, for a total of 16 combinations of shifts. Each combination was presented for 110 s twice in interleaved format with movements happening every 0.5 s. The first 10 s of each presentation were discarded leaving 200 trials per condition, with an equal number of left and right shifts that were analyzed independently (see *Figure 1C*).

### Linear-nonlinear model
LN models for Gaussian stimuli (*Figure 3*) consisted of the light intensity passed through a linear temporal filter, which describes the average response to a brief flash of light in a linear system, followed by a static nonlinearity, which describes the threshold and sensitivity of the cell (*Baccus and Meister, 2002*). To compute LN models for white noise stimuli, we first computed linear filters, $F(t)$, which were the time-reverse of the spike-triggered average. Then, we calculated linear prediction, $g(t)$, as the convolution of the temporal filter and the central stimulus, $s(t)$,

$$g(t) = \int F(\tau)s(t-\tau)d\tau \qquad (2)$$

A static nonlinearity, $N(g)$, was computed by averaging the value of the firing rate, $r(t)$, over bins of $g(t)$. The filter, $F(t)$, was normalized in amplitude such that it did not amplify the stimulus, i.e. the variance of $s$ and $g$ were equal (*Baccus and Meister, 2002*). Thus, the linear filter contained only relative temporal sensitivity, and the nonlinearity represented the overall sensitivity of the transformation.

For pink noise stimuli (*Figure 2*), a sequence was generated with an amplitude spectrum that was inversely proportional to the frequency ($1/f$). Because the stimulus contained temporal correlations, the linear filter was computed by reverse correlation while normalizing by the autocorrelation of the stimulus (*Baccus and Meister, 2002*).

### Mutual information as a function of time
In our experimental designs, the full stimulus $S$ consisted of two components, the periphery, $P$, and a center stimulus $C$. In all experiments, the periphery was either still (with zero entropy and thus the set of responses $R$ contained no information about it), or reversed at 1 Hz. By discretizing time in 50 ms bins we create 20 different peripheral conditions $p \in P$, each of which represents a time relative to the period of the peripheral stimulus. By the chain rule of information (*Cover and Thomas, 1991*)

$$I(R;P,C) = I(R;P) + I(R;C|P) \qquad (3)$$

The last term can be understood as the average information that the response carries about the central stimulus if the peripheral stimulus was known. This equation can also be written as (*Cover and Thomas, 1991*)

$$I(R; P, C) = I(R; P) + \langle I(R; C|p) \rangle_{p \in P} \tag{4}$$

where the last term is an average over all instances of the peripheral stimulus $p \in P$.

We computed $I(R; C|p)$ (the quantity inside the $\langle . \rangle_{p \in P}$ in eq (8)) for each time bin $p$ relative to the peripheral period (*Figure 2D*, inset) so that for each $p$ there is no information between the cell's response and the peripheral stimulus (because under the set of stimuli analyzed there is only one peripheral stimulus, which has zero entropy). To analyze how information varies as a function of time relative to the peripheral shift, we show $I(R; C|p)$ averaged over both phases of the periphery, each of which had a similar time course (*Figure 2D*, inset).

## Pink noise analysis

A 200 s sequence of a Gaussian pink noise ($1/f$ amplitude spectrum) stimulus with an equivalent contrast of 10% was repeated 10–20 times. For the stability of information calculations with this number of repeats, see (*Figure 2—figure supplement 1B*). In the Still condition, the periphery was static and in the Shift condition the peripheral checkerboard shifted every 0.5 s. As stated above, each time relative to the peripheral period was analyzed separately, so that under each condition there was no information between the response of a cell and the periphery's position. Although the central sequences were not the same for each time bin relative to the peripheral shift, by having 200 central sequences per periphery we limit the chance of biasing any particular periphery by associating it with more (or less) discriminable central stimuli. Center sequences were identical between the Still and the Shift conditions, and therefore the differences between $I(R; C|p)$ under still and shift for any peripheral stimulus $p$ is only attributable to the one difference in the experimental conditions, the presence or absence of peripheral stimulation. The response, $r_i$, during trial $i$ was defined as the number of spikes in a 50 ms time bin; other intervals from 12 to 160 ms yielded similar results (*Figure 2—figure supplement 1B*).

The mutual information, $I(R; C|p)$, was computed by taking the difference between the total response entropy, $H(R|p)$, and the conditional (noise) entropy, $H(R; C|p)$ (*Cover and Thomas, 1991*),

$$I(R; C|p) = H(R|p) - H(R|C, p) \tag{5}$$

where

$$H(X_1|X_2) = -\sum_{x_1, x_2} p(x_1, x_2) \log_2 \Big( p(x_1|x_2) \Big) \tag{6}$$

Entropy values were calculated from a histogram estimate of probability distributions.

## Dynamics of contrast and sequence information

The central stimulus followed a 4-bit *M*-sequence, with each stimulus frame having one of two values, $\mu + \Delta I$ and $\mu - \Delta I$, and a Michelson contrast ($\Delta I / \mu$) of one of four possible values, 3, 6, 12 and 24%. Each four frame sequence occurred once in a repetition of the *M*-sequence, where $M^{(4)}$ is the set of all 16 possible combinations of four binary digits. The luminance in the center was updated at 10 Hz, and therefore one presentation of the *M*-sequence lasted $0.1s \cdot 2^4 = 1.6s$. The sequence was repeated for 11 trials at a given contrast and the responses for the first trial after a transition to a new contrast were discarded from the analysis. Contrasts were picked randomly without replacement and then a different order chosen once all four contrasts were tested. A stimulus was defined as the combination of the center (both sequence and contrast) and the periphery. By binning time in 100 ms there are 10 possible peripheral stimuli (time $p$ relative to peripheral period), 16 possible sequences, $m^{(4)}$ and 4 possible contrasts (σ) for a total of 640 different stimuli. Each stimulus was measured at least 10 times.

When the central stimulus is divided into the stimulus sequence, $m^{(4)} \in M^{(4)}$ and the contrast, $\sigma \in \sum$, the total information from eq (1) can be further expanded into:

$$I(R;P,\sum,M^{(4)}) = I(R;P) + I(R;\sum|P) + I(R;M^{(4)}|\sum,P)$$
$$= I(R;P) + \left\langle I(R;\sum|p) + I(R;M^{(4)}|\sum,p)\right\rangle_{p\in P} \tag{7}$$

Where $I(R;\sum|p)$ and $I(R;M^{(4)}|\sum,p)$ are functions of time $p$. The quantity $I(R;M^{(4)}|\sum,p)$ represents the information the brain could extract about the stimulus-sequence at a given time relative to the peripheral stimulus if it knew the contrast. Whereas $I(R;M^{(4)}|p)$ represents the information that the brain could extract about the stimulus sequence at a given time if it did not know the contrast. Comparisons of the time course of $I(R;M^{(4)}|p)$ and $I(R;\sum|p)$ (**Figure 4C**), as well as between $I(R;M^{(4)}|\sum,p)$ and $I(R;\sum|p)$ (**Figure 4—figure supplement 1**), indicate that the dynamics of information about sequence and contrast are different whether the brain can use contrast information to decode the sequence or not.

Spatial extent of peripheral changes in sensitivity.

To measure the spatial extent of increases and decreases in sensitivity, the central stimulus consisted of a mask in the pattern of a checkerboard, with squares 100 μm in size, that flickered with a pink noise stimulus intensity that was the same in all squares (**Figure 5A**). The overall size of the mask was 1.2 mm. The central stimulus was identical in all conditions. The background was a more finely scaled checkerboard, with squares 50 μm in size, and a central blank region that was varied in size from the full size of the monitor (no checkers in the periphery) to 0 μm (checkers everywhere except in central stimulus). At smaller values of the central blank region, the background was intercalated with the central region (**Figure 5B**, bottom). For each location of the peripheral stimulus, we computed an LN model between the center stimulus and the cell's response and calculated the average slope of the nonlinearity for different time windows.

## Model integrating central and peripheral input

The model of long-range excitation consisted of a central stimulus $s(t)$ that was passed through a linear filter $F(\tau)$, yielding the filtered central input

$$b(t) = \int_0^1 F(\tau)s(t-\tau)d\tau \tag{8}$$

The filtered central input was combined with a signal, $a(t)$, that depended on background motion. Because the goal of this model was to investigate the integration of central and peripheral signals, and not the origin of the peripheral signal as has been studied elsewhere in greater detail (**Passaglia et al., 2009**), we defined $a(t)$ to be a biphasic function of time that began at the time t = 0, representing the time of the saccade. As to a plausible origin of this input, the peripheral effect occurred for a high spatial frequency checker and did not depend on the direction of the shift. Such a response could be generated by a group of rectified subunits as found in the receptive fields of polyaxonal amacrine cells (**Baccus et al., 2008**), but this was not explicitly implemented here. Because $a(t)$ had two phases, positive and negative, whereas polyaxonal amacrine cells are thought only to deliver inhibition, it is expected that the positive phase would arise through disinhibition delivered through a second intervening amacrine cell that provides tonic inhibition.

The central and peripheral inputs were then passed through a threshold nonlinearity $N(.)$.

$$c(t) = N(b(t) + a(t)) \tag{9}$$

The nonlinearity was chosen with a slope of one and the threshold equal to 0.9 times the maximum amplitude of the peripheral signal $a(t)$·.

The output of the threshold function was then scaled by feedforward divisive adaptation, yielding the model output

$$y(t) = \frac{c(t)}{\alpha + \int F_\alpha(\tau)c(t-\tau)d\tau} \tag{10}$$

$F\alpha(t)$ was an exponentially decaying filter with an integral of one, and a time constant set to 10 s, although this parameter could be varied from 1 to 100 s with little effect. Smaller values of $\alpha$ yield more complete adaptation. However, for constant luminance experiments, equation (10) reduces to

a constant independent of the luminance value when $\alpha = 0$, therefore non-zero values of α are needed. Thus, the value of alpha was optimized to yield changes in adaptation as observed in the data, as well as responses to different levels of constant luminance. We used $\alpha = 0.30$, but values from 0.05 to 2.00 could be used with similar results.

## Spatiotemporal model under global shifts

From previous studies, the first sharp threshold encountered in the retina is at the bipolar cell synaptic terminal (*Mennerick and Matthews, 1996*). Thus, the filter in the model should correspond to the spatiotemporal receptive field of a bipolar cell, which we took from our previous measurements of fast-Off bipolar cells (*Baccus et al., 2008*) (*Figure 7A*). In order to use the model to assess information transmission, we measured the noise in the membrane potential of bipolar cells as a function of contrast and found that the level of noise increases roughly linearly with contrast (*Figure 7A-iii*). This allowed us to choose a noise level at each time point that depended on the filtered stimulus, approximating measured bipolar cell noise.

A set of 342 images were taken from a database of natural scenes (*Tkačik et al., 2011*). The bipolar cell membrane potential was simulated by combining a linear receptive field pathway (*Figure 7A-i*) and noise (*Figure 7A-iii*). Because the filtering and noise of bipolar cells was measured at a fixed mean intensity, to avoid the need to incorporate luminance adaptation into the model, we normalized the mean intensity of all images. The linear receptive field center and surround were modeled independently, with each being a filter separable in space and time. Spatial linear predictions were made by convolving each image with a spatial disk of 1.0 and 2.5 degrees of visual space corresponding to center and surround. To compute the complete linear prediction of cells, images were jittered around according to a random walk with mean velocity of 0.33° per second simulating fixational eye movements and an instantaneous saccade simulated by a step of 6° in a random orientation in the image location. The temporal receptive fields for the center and surround were convolved with this image series and summed, generating the complete linear prediction for each location as a function of time; $g_{x,t}$ where $x$ denotes the location and $t$ the time. From the 342 images, a total of 82,863 identical bipolar cells with non-overlapping centers were simulated.

Because bipolar cell noise depends on the stimulus contrast (*Figure 7A-iii*), we used a model of the noise whereby the instantaneous standard deviation of the noise at each point in time relative to the shift depended on the standard deviation of the linear prediction across the bipolar cell population (*Figure 7A-iii*, lower panel, black trace). This created the greatest noise during the gating window, and thus potentially underestimates the actual information conveyed. From the signal standard deviation at a particular time, an equivalent Gaussian contrast was found that would generate a linear prediction with the same standard deviation. With the equivalent Gaussian contrast, a level of noise was chosen such that the signal to noise ratio was the same in the simulation and in the bipolar cell's measured membrane potential noise under repetitions of Gaussian stimulation at different contrasts (*Figure 7A-iii*, upper panel). Each cell in the model received independent noise which was generated from a Gaussian distribution.

## Information calculations from the model

The linear prediction $g(t)$ and model output $r(t)$ were binned to compute mutual information and conditional mutual information. The linear prediction $g(t)$ and the response $r(t)$ were divided into 16 unequal bins, positioned to maximize information about the total range of $g(t)$ and $r(t)$. The same bins were kept for all delays, $p$ relative to the shift. The information that the response carries about the linear prediction at a particular delay $p$ relative to the shift, was computed as $I(G_t; R_t|p)$ by taking the difference between the total response entropy at a given delay, $H(R_t|p)$, and the conditional (noise) entropy, $H(R_t|G_t, p)$. The conditional mutual information between the linear prediction $g(t)$ and the response $r(t)$ given the previous response $r(t - \Delta)$ at a given delay, $p$, was computed as

$$I(G_t; R_t|R_{t-\Delta}, p) = H(G_t|R_{t-\Delta}, p) - H(G_t|R_{t-\Delta}, R_t, p) \tag{11}$$

## Adaptation index

To compute a change in adaptation at different times relative to a shift, an index was computed that compared the measured change in the slope of the nonlinearity to the change expected from

complete adaptation. After presenting a series of contrasts $\sigma_i$, we computed the nonlinearities $N(.)$ of an LN model, and from each nonlinearity extracted the slope $m_i$. Picking one contrast as reference, we normalized the slopes and the contrasts by those of the reference $\tilde{m} = m/m_0$ and $\bar{\sigma} = \sigma/\sigma_0$ and fitted a line to $\tilde{m}$ vs. $1/\sigma$. The adaptive index is the slope of the fitted line, which will be one for an ideally adapting cell and zero for a nonadapting cell (*Figure 3—figure supplement 1A–B*).

### Ideal information

The goal was to find the nonlinearity that maximized the mutual information $I(G; R)$, between the set of linear predictions, $G$ and the set of spike counts, $R$, given the noise properties of the cell. We began with the linear prediction as a function of time $g(t)$, the spike count distribution at that time, $P\big(r(t)\big)$, computed over trials, and the average rate at that time, $\langle r(t) \rangle$, computed by averaging over trials. The nonlinearity $N_0(g)$ maps $g(t)$ at a time $t$ onto a model average firing rate $\langle r'(t) \rangle$ at that time, but to include noise that was most consistent with the observed noise we computed from the data the distribution of spike counts for a given average rate, $P\big(r(t)|\langle r \rangle\big)$. This function mapped each average rate $\langle r(t) \rangle$ at each time onto a distribution $P\big(r(t)\big)$. The optimized nonlinearity, $N_0(g)$, was a sigmoid parameterized by a slope, $x_1^{-1}$ and a midpoint, $x_0$, and was constrained to have the same minimum (zero) and maximum rate as the measured data $N_0(x) = \frac{r_{max}}{1+e^{-(x-x_0)/x_1}}$. To find for each candidate $N_0(.)$ the best estimated joint distribution $P\big(g(t), r'(t)\big)$ between $g(t)$ and the model distribution of spike counts $r'(t)$, we used $N_0(.)$ to map $g(t)$ onto $\langle r'(t) \rangle$, and then used the function $P\big(r(t)|\langle r \rangle\big)$ to map $\langle r \rangle$ onto $P\big(r'(t)|g(t)\big)$ for a particular value of $g(t)$, i.e. $P\big(r'(t)|g(t)\big) = P\big(r(t)|N_0(g)\big)$. Then, we weighted this conditional distribution by the marginal probability of the linear prediction $g$, $P(g)$, which has a Gaussian distribution, to compute the full joint distribution of $g(t)$ and $r'(t)$,

$$P\big(g(t), r'(t)\big) = P\big(g(t)\big) P\big(r'(t)|g(t)\big) \tag{12}$$

from which the mutual information was computed. Then we performed a grid search of the parameters of $N_0(\ )$ and found from $P\big(g(t), r'(t)\big)$ the nonlinearity that maximized $I(G_t, R_t')$. The maximum value of $I(G_t, R_t')$ was taken as the maximum amount of information given the measured noise of the cell, and its minimum and maximum firing rate.

## Acknowledgements

We thank D Kastner, S Ganguli, T Clandinin and L Nevin for comments on the manuscript; T Moore and D Kastner for helpful discussions. This work was supported by grants from the US National Eye Institute, Pew Charitable Trusts, McKnight Endownment Fund for Neuroscience, E. Mathilda Ziegler Foundation (SAB); and by a Walter & Idun Berry Fellowship (PDJ).

## Additional information

### Funding

| Funder | Grant reference number | Author |
| --- | --- | --- |
| Walter & Idun Berry Foundation | Walter & Idun Berry Fellowship | Pablo D Jadzinsky |
| McKnight Endowment Fund for Neuroscience | McKnight Scholar Award | Stephen A Baccus |
| E. Mathilda Ziegler Foundation | | Stephen A Baccus |
| Pew Charitable Trusts | Pew Scholarship in the Biomedical Sciences | Stephen A Baccus |

| National Eye Institute | R01EY16842 | Stephen A Baccus |
| National Eye Institute | R01EY022933 | Stephen A Baccus |

The funders had no role in study design, data collection and interpretation, or the decision to submit the work for publication.

## Author contributions

PDJ, designed the study, performed the experiments and analysis, wrote the manuscript, Conception and design, Acquisition of data, Analysis and interpretation of data, Drafting or revising the article; SAB, designed the study, wrote the manuscript, Conception and design, Drafting or revising the article

## Ethics

Animal experimentation: This study was performed in strict accordance with the recommendations in the Guide for the Care and Use of Laboratory Animals of the National Institutes of Health, and the Stanford institutional animal care and use committee (IACUC) protocol (11619).

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
