## [Decision Letter]

Thank you for submitting your work entitled "Synchronized amplification of local information transmission by peripheral retinal input" for peer review at *eLife*. Your submission has been favorably evaluated by Eve Marder (Senior editor) and 3 reviewers, one of whom, Ronald L. Calabrese, is a member of our Board of Reviewing Editors.

The reviewers have discussed the reviews with one another and the Reviewing editor has drafted this decision to help you prepare a revised submission.

Summary:

The authors present an electrophysiological and modeling analysis of how local information transmission is gated by peripheral input in salamander retina. This analysis is currently framed in terms of how eye movements like those in saccades influence local signal transmission. By using clever stimulus protocols and multielectrode recordings they show that rapid shifts in peripheral input can gate responses in the center and locally amplify signal transmission to increase information flow. Modeling employing LNL models serves to corroborate and amplify the experimental findings and provide an explanation of potential mechanism. Underlying this gating of information is a transient increase in adaptation to contrast, enhancing sensitivity to a broader range of stimuli. They interpret their results as indicating a dynamic tradeoff between energy conservation and maximizing information flow. The retina switches from a coding strategy that conserves energy to one of increased information transmission at the expected time of higher information content that occurs after an eye movement.

Essential revisions:

While the basic finding that rapid shifts in peripheral input can gate responses in the center is novel, interesting, and well supported by the data there are two concerns that must be addressed by the authors before this paper can be considered for publication in *eLife*.

1) The analogy of the peripheral stimuli to saccades is an over-reach. The animals do not have eye movements but rather head sweeps and these are not defined as to size, speed, or frequency. In the absence of testing stimulus specificity of the effect, and whether that matches expectations from head movements in salamander, you might emphasize the strong and unexpected effects of stimuli in the far surround on coding. Such a change in emphasis would make the paper stronger more generally applicable.

2) The LNL model depends heavily on the assumption of linearity in the bipolar cell responses (almost certainly an approximation as photoreceptor adaptation to luminance would place at least some adaptation prior to the combination of center and peripheral inputs), which in turn depends on the analysis of the noise data in Figure 7. The conditions for these experiments are discussed only briefly in the Methods and need to be fully explained. Noise measurements are tricky since there are so many confounding factors (e.g., drift in the recording, instrumental noise, etc.) and more information about the analysis is needed to evaluate that data critically.

The comments of the expert reviewers provide further details for addressing these two main concerns.

Reviewer #1:

The authors present an electrophysiological and modeling analysis of how local information transmission is gated by peripheral input in salamander retina. This analysis is framed in terms of how eye movements like those in saccades influence local signal transmission. By using clever stimulus protocols and multielectrode recordings they show that simulated movements locally amplify signal transmission to increase information flow. Modeling employing LNL models serves to corroborate and amplify the experimental findings and provide an explanation of potential mechanism. Underlying this gating of information is a transient increase in adaptation to contrast, enhancing sensitivity to a broader range of stimuli. They interpret their results as indicating a dynamic tradeoff between energy conservation and maximizing information flow. The retina switches from a coding strategy that conserves energy to one of increased information transmission at the expected time of higher information content that occurs after an eye movement.

These experiments seem carefully done and the data presented is important and solid and supports the conclusions. The writing is in general clear; the authors have made the effort to reach out to the general reader but there are places where more explanation of the rather complicated analysis and modeling would help. Methods are adequately presented.

Reviewer #2:

This is an interesting paper that makes a number of observations about how stimuli in the far periphery of a ganglion cell's receptive field influence coding by the receptive field center. The central result in the paper – that rapid shifts in peripheral input can gate responses in the center – is well supported by the data. I found some other aspects less complete, as detailed below. Comments are in approximate order of importance.

1) The paper is motivated and the results interpreted in the context of saccades. I did not find this convincing. First, it is not clear how well the sudden spatial shift of the peripheral stimulus replicates the types of eye/head movements made by salamanders. Some key features – such as coherent motion across the retina (and receptive field) – are certainly missing. Second, the importance of peripheral stimuli for coding depends on the frequency of such events. How often do salamanders make rapid head movements? These issues are particularly important given the underlying theme in the paper that the interaction between peripheral and central stimuli serves to create an energetically efficient neural code (Abstract, first paragraph of Introduction). An alternative, framework for the paper is the literature on how non-classical surround stimulation alters responses.

2) A second major issue is the accuracy and completeness of the proposed model. The evidence for the model is quite indirect, and it is not clear whether the model is unique in its ability to capture the data. This raises concerns about how well the model will capture a broad range of inputs – particularly inputs not directly tested. Because of this issue, I found the section of the paper describing responses to natural stimuli tenuous. A specific concern about the model is whether adaptation is located appropriately in the model (e.g. photoreceptors adapt strongly to luminance, which seems important to account for). The location of adaptation relative to the combination of peripheral and central responses is critical to the model (as pointed out in the paper), and hence needs to be tested more completely. Another concern is the assumption that the bipolar cell responses to natural inputs are linear (and can be captured by a simple impulse response). This should be tested. A third concern is how the eye movement trajectories were generated, and whether those are appropriate for salamander (see above). Given these issues I think the model results need to be discussed more tentatively or qualified as being subject to the caveats associated with the model.

3) A third major issue is a lack of needed details for interpreting some aspects of the paper. One example is the measurements underlying the bipolar component of the model described above. Another example is the subheading “Peripheral stimulation gates a change in adaptation”, where the statements in the text about key conclusions need to be tied more closely to the results.

*Reviewer #3:*

This paper describes an analysis of the effects of peripheral "global motion", such as might be induced by a saccade, on the response of ganglion cells in salamander retina to the stimuli within their local receptive fields. It shows such shifts have a dramatic effect on local processing in a short time window after the shift:

a) Spike rate is increasedb) Information transmission is amplifiedc) Adaptation becomes more complete (i.e., so information about the contrast is thrown away more effectively).

The paper describes a (phenomenological) adapting LN model that can account for these effects very accurately, and perform a nice analysis of the effects of this mechanism on the processing of natural scenes during natural vision. The paper is well written, the analyses are thorough and the findings are extremely interesting. I have a few concerns and questions but overall I think it makes a very nice contribution to the literature.

Comments:

1) I have one technical concern about the effects of the classical RF surround on these analyses. Early on the paper the authors say that "the object covered the classical linear receptive field center plus part of the surround of most cells", suggesting that perhaps a majority of the cells would have some portion of the peripheral checkerboard pattern within their classical RF surround. There are several analyses showing that similar effects of left and right shifts (which would be expected to have opposite effects on a linear surround) – this seems to be the main form of evidence that we're not seeing classical RF surround effects. But I don't know enough about salamander retina to know whether this is reasonable or not. For example, do we know definitively that these cells don't have more complex cat Y-like surrounds?

The evidence presented later in Figure 6 would seem to allay these concerns, due to the fact that the effect extends out to 1mm (20 degree eccentricity) from the RF, which is presumably far outside the classical RF. But it would be helpful to see some numbers here about the size of the typical RF center and surround for these neurons. Mostly I'd like to see the authors make a slightly more careful distinction between effects we might expect from the RF surround vs. those we're seeing here. (Or, conversely, if the RF surround contributes strongly to these effects, overlaid with some additional long-range mechanisms, it would be worth knowing that as well).

2) The manipulations of the surround seem to all involve shifts, such as one might encounter with saccadic eye movements. But I'm curious to know if the authors ever examined other kinds of changes to the peripheral stimulus, e.g., a change of the spatial pattern that doesn't look anything like a motion stimulus. In other words, is a big change in the surround stimulus sufficient to give rise to these effects, or does it specifically require motion? I wouldn't consider this necessary for publication if the authors haven't gathered any such data, but if they have reason to argue for the specificity of motion "shift" stimuli, I'd like to hear it. (I apologize if the paper already says this somewhere and I simply missed it.)

3) I'm a little bit confused about how the analysis of information efficiency was carried out. (Is this exactly the same as the analysis from the Pitkow and Meister paper?)

The authors state: "Our analysis indicates that after a peripheral shift, the neural code shifts away from energy conservation and towards high- throughput information transmission." This seems a bit odd though – before the shift, a cell is at 24% of its maximum achievable information, then after a shift the cell is achieving 47% of its theoretical maximum information transmission, meaning it become more efficient while also increasing spike rate? So that doesn't sound like a tradeoff. I was expecting it to achieve higher total information flow but at lower efficiency. Can you speculate about why it's efficient at high spike rates and inefficient at low spike rates? I thought the high threshold argument meant that the opposite should occur (use a high threshold and be highly efficient while emitting few spikes).

This seems to directly go against the water-filling argument in the Discussion: the analogy there would be that you use the most efficient parts of the channel first (where you get the highest amount of information per spike possible), and only when you need to cram more bits through the channel do you need to use the less efficient parts. So what's going on here?

4) Why don't we see effects of peripheral inhibition, i.e., those putatively responsible for saccadic suppression?

[Editors' note: further revisions were requested prior to acceptance, as described below.]

Thank you for resubmitting your work entitled "Synchronized amplification of local information transmission by peripheral retinal input" for further consideration at *eLife*. Your revised article has been favorably evaluated by Eve Marder (Senior editor) and two reviewers, one of whom is a member of our Board of Reviewing Editors. The manuscript has been improved but there are some remaining issues that need to be addressed before acceptance, as outlined below:

The authors have made a systematic and cooperative effort to answer the concerns of the previous review. Specifically they have addressed several technical aspects and they have refocused the paper from the study of eye movements to the more general study of how the retina adjusts its response to the expected statistics of the sensory input.

The supplementary data presented are helpful and the expanded Materials and methods section is appropriate. The writing is in general clear; the authors have made the Discussion very clear and accessible.

Nevertheless there remains a significant concern that must be addressed.

A key feature of the model structure is that the bipolar responses are linear and hence that the peripheral input occurs prior to the nonlinearity in the model. The new data added to the paper on bipolar responses do not make a convincing case for linearity of the bipolar responses to natural stimuli. Specifically, while the bipolar responses appear close to linear for 20% contrast noise stimuli (as in new figure), the natural scene stimuli seem very likely to involve much larger changes in input (typical scenes include regions that differ 20-50 fold in intensity). Hence the model involves extrapolation of the bipolar responses well beyond the range where linearity was tested. As noted in the previous round of reviews, photoreceptors are known to adapt strongly to changes in their mean input. It is not clear from the paper what the statistics of the natural inputs are for the model in Figure 7, but it would be surprising if the photoreceptor responses were linear across these stimuli. Published work supports nonlinearities in the photoreceptor responses to natural inputs (Endeman and Kamermans, 2010). Normalizing the mean of the entire image is not adequate to prevent adaptation in the photoreceptors since that process will not change the fact that separate regions differ substantially in intensity, and dynamic adaptation will occur during movement of the image on the retina.

The manuscript should be revised either by directly testing the model assumptions, if the authors have those data available, or shifting the model into the Discussion and treating in more like a 'toy' or heuristic model with the assumptions that the model is based on (and caveats to those assumptions) stated clearly.

---

## [Author Response]

Essential revisions:

*While the basic finding that rapid shifts in peripheral input can gate responses in the center is novel, interesting, and well supported by the data there are two concerns that must be addressed by the authors before this paper can be considered for publication in* eLife.

1) The analogy of the peripheral stimuli to saccades is an over-reach. The animals do not have eye movements but rather head sweeps and these are not defined as to size, speed, or frequency. In the absence of testing stimulus specificity of the effect, and whether that matches expectations from head movements in salamander, you might emphasize the strong and unexpected effects of stimuli in the far surround on coding. Such a change in emphasis would make the paper stronger more generally applicable.

We have shifted the context of the paper from the study of eye movements to the more general study of how the retina adjusts its response to the expected statistics of the sensory input. Motion in the periphery – whether it be from a large moving object, an eye or head movement – generates an expectation that the stimulus statistics in the center will change, suddenly providing novel information. We have removed references to saccades in the Results, except for the model in Figure 7, where we say that the global shift in the image could represent an eye movement. In the Discussion, because long-range excitation has been seen in many species including primates, we still raise the issue that these effects are expected to be relevant to eye movements, including saccades.

*2) The LNL model depends heavily on the assumption of linearity in the bipolar cell responses (almost certainly an approximation as photoreceptor adaptation to luminance would place at least some adaptation prior to the combination of center and peripheral inputs), which in turn depends on the analysis of the noise data in Figure 7. The conditions for these experiments are discussed only briefly in the Methods and need to be fully explained. Noise measurements are tricky since there are so many confounding factors (e.g., drift in the recording, instrumental noise, etc.) and more information about the analysis is needed to evaluate that data critically.*

We have now added a supplemental figure that describes how the bipolar cell noise was measured. Fast off bipolar cells are roughly linear at a constant mean intensity (Figure 7—figure supplement 1). Furthermore, these bipolar cells are appropriate to use for these particular ganglion cells, as a model composed of fast Off bipolar cell linear filters followed by rectification has been shown to accurately capture the responses of fast Off-type ganglion cells to a spatiotemporal stimulus, namely a grating jittering with fixational eye movements (Baccus et al., 2008). Although these bipolar cells will not be linear if the mean intensity changes, the filtering and noise of bipolar cells was measured at a fixed mean intensity. Furthermore, in the simulation to avoid incorporation luminance adaptation into the model, we normalized the mean intensity of all images. We give further details as to how noise measurements in bipolar cells were conducted by repeating random flicker at a fixed mean intensity (Figure 7—figure supplement 1). In addition, we repeat the information calculation while changing the noise by a factor of two, showing that the results change little qualitatively (Figure 7—figure supplement 2).

Reviewer #2:

This is an interesting paper that makes a number of observations about how stimuli in the far periphery of a ganglion cell's receptive field influence coding by the receptive field center. The central result in the paper – that rapid shifts in peripheral input can gate responses in the center – is well supported by the data. I found some other aspects less complete, as detailed below. Comments are in approximate order of importance.

1) The paper is motivated and the results interpreted in the context of saccades. I did not find this convincing. First, it is not clear how well the sudden spatial shift of the peripheral stimulus replicates the types of eye/head movements made by salamanders. Some key features – such as coherent motion across the retina (and receptive field) – are certainly missing. Second, the importance of peripheral stimuli for coding depends on the frequency of such events. How often do salamanders make rapid head movements? These issues are particularly important given the underlying theme in the paper that the interaction between peripheral and central stimuli serves to create an energetically efficient neural code (Abstract, first paragraph of Introduction). An alternative, framework for the paper is the literature on how non-classical surround stimulation alters responses.

We have reframed the paper in terms of the change in neural coding in the receptive field center and its relation to the expected change in statistics based on measurements in the periphery. This topic includes potential effects from eye movements, but we have limited the potential relationship to saccadic eye movements to a single paragraph in the Discussion.

*2) A second major issue is the accuracy and completeness of the proposed model. The evidence for the model is quite indirect, and it is not clear whether the model is unique in its ability to capture the data.*

The general issue of uniqueness would be more of an issue in a less well studied system, but the computational structure of the excitatory pathways of ganglion cells has received substantial previous attention, and thus our model is constrained by prior work. We have now modified the main text (subheading “A simple model produces gating and changes in adaptation”) to point out that based on prior work the only decision in constructing the model was the level at which to combine peripheral input, and then we explain why based on our data why this level has to be before the nonlinearity.

*This raises concerns about how well the model will capture a broad range of inputs – particularly inputs not directly tested.*

It is true that this model may not accurately predict the response to natural scenes of ganglion cells because of other nonlinear properties of the retina not represented here, but the model is effectively of the release from a single bipolar cell. Other nonlinear properties in the inner retina may occur subsequent to the output stage of the model we use. In addition, we have shifted the focus to investigate the timing of new information as filtered through the linear filtering properties of the receptive field center and the timing of peripheral input. Thus we think the model sufficiently constrained by our measurements and the literature, and it is used in an appropriate way to extrapolate in a reasonable way beyond currently feasible experiments. We now point this out in the subsection “A model compares the timing of gating with the expected increase in information after a global shift”.

Because of this issue, I found the section of the paper describing responses to natural stimuli tenuous. A specific concern about the model is whether adaptation is located appropriately in the model (e.g. photoreceptors adapt strongly to luminance, which seems important to account for). The location of adaptation relative to the combination of peripheral and central responses is critical to the model (as pointed out in the paper), and hence needs to be tested more completely.

We believe that the experiments we have done are sufficient to rule out that the peripheral input is integrated after the nonlinearity, which would produce a vertical shift in the nonlinearity of an LN model fit to the whole response, and we can also rule out that the peripheral input is integrated after adaptation, as would be the case with a postsynaptic effect, because this effect would apply to both low and high contrast central input.

*Another concern is the assumption that the bipolar cell responses to natural inputs are linear (and can be captured by a simple impulse response). This should be tested.*

Previous experiments for fast Off bipolar cells have shown their responses are roughly linear for spatiotemporal stimuli for a fixed mean stimulus (Baccus et al., 2008). We also now include data comparing a linear model to the bipolar cell response for a Gaussian white noise uniform field stimulus (Figure 7—figure supplement 1).

A third concern is how the eye movement trajectories were generated, and whether those are appropriate for salamander (see above). Given these issues I think the model results need to be discussed more tentatively or qualified as being subject to the caveats associated with the model.

Although we have reduced the overall connection with eye movements, we include a paragraph in the Discussion (subheading “Potential role of peripheral gating during eye movements”) addressing the differences between salamander and mammal eye movements. Furthermore, we point out that an important conclusion derived from the model is that the timing of the linear filter in the receptive field center is coincident with temporal filtering from the peripheral input. Thus even if the global shift is more smooth as in the case of a larger mammalian saccade or an amphibian head saccade, we expect that the excitation from both center and periphery will still coincide.

3) A third major issue is a lack of needed details for interpreting some aspects of the paper. One example is the measurements underlying the bipolar component of the model described above. Another example is the subheading “Peripheral stimulation gates a change in adaptation”, where the statements in the text about key conclusions need to be tied more closely to the results.

We now include a supplemental figure (Figure 7—figure supplement 2) showing how noise in bipolar cells was measured, as well as references describing the constraints of spatiotemporal filtering, threshold and adaptation already present from previous literature.

Reviewer #3:

*[…] The paper describes a (phenomenological) adapting LN model that can account for these effects very accurately, and perform a nice analysis of the effects of this mechanism on the processing of natural scenes during natural vision. The paper is well written, the analyses are thorough and the findings are extremely interesting. I have a few concerns and questions but overall I think it makes a very nice contribution to the literature.*

*Comments:*

1) I have one technical concern about the effects of the classical RF surround on these analyses. Early on the paper, the authors say that "the object covered the classical linear receptive field center plus part of the surround of most cells", suggesting that perhaps a majority of the cells would have some portion of the peripheral checkerboard pattern within their classical RF surround. There are several analyses showing that similar effects of left and right shifts (which would be expected to have opposite effects on a linear surround) – this seems to be the main form of evidence that we're not seeing classical RF surround effects. But I don't know enough about salamander retina to know whether this is reasonable or not. For example, do we know definitively that these cells don't have more complex cat Y-like surrounds? The evidence presented later in Figure 6 would seem to allay these concerns, due to the fact that the effect extends out to 1mm (20 degree eccentricity) from the RF, which is presumably far outside the classical RF. But it would be helpful to see some numbers here about the size of the typical RF center and surround for these neurons. Mostly I'd like to see the authors make a slightly more careful distinction between effects we might expect from the RF surround vs. those we're seeing here. (Or, conversely, if the RF surround contributes strongly to these effects, overlaid with some additional long-range mechanisms, it would be worth knowing that as well).

To clarify, in Figure 1 we are claiming that peripheral excitation is not contributed by the linear receptive field surround because shifts of the opposite sign produce the same effect – a nonlinear property. Even if the nonlinear effect did arise from the spatial region occupied by the linear receptive field surround or even center, this nonlinear effect must arise from a pathway distinct from the ones that generate the linear receptive field. The evidence on Figure 6 further supports this, because a fine grating at such a distance would not activate the linear receptive field surround, even once again if the surround extended to this distance. We clarify these two points in paragraph four of the subsection “During global shifts, peripheral stimulation increases the response to local stimuli”, and in “Peripheral excitation and inhibition act across a similar spatial scale”.

2) The manipulations of the surround seem to all involve shifts, such as one might encounter with saccadic eye movements. But I'm curious to know if the authors ever examined other kinds of changes to the peripheral stimulus, e.g., a change of the spatial pattern that doesn't look anything like a motion stimulus. In other words, is a big change in the surround stimulus sufficient to give rise to these effects, or does it specifically require motion? I wouldn't consider this necessary for publication if the authors haven't gathered any such data, but if they have reason to argue for the specificity of motion "shift" stimuli, I'd like to hear it. (I apologize if the paper already says this somewhere and I simply missed it.)

We did not use other peripheral stimuli, but now include a comment that due to the similar timing of linear filtering from the center and gating, we would expect that with more smooth stimuli the effects would still coincide (paragraph three, subsection “Potential role of peripheral gating during eye movements”).

3) I'm a little bit confused about how the analysis of information efficiency was carried out. (Is this exactly the same as the analysis from the Pitkow and Meister paper?)

It is a bit different, in that the average firing rate was not constrained, rather the maximum of the nonlinearity was kept constant, so that a leftward shift would result in an increase in firing. In addition, we used the actual noise distribution instead of a model of the noise. We now give more details in the Materials and methods.

The authors state: "Our analysis indicates that after a peripheral shift, the neural code shifts away from energy conservation and towards high- throughput information transmission." This seems a bit odd though – before the shift, a cell is at 24% of its maximum achievable information, then after a shift the cell is achieving 47% of its theoretical maximum information transmission, meaning it become more efficient while also increasing spike rate? So that doesn't sound like a tradeoff. I was expecting it to achieve higher total information flow but at lower efficiency. Can you speculate about why it's efficient at high spike rates and inefficient at low spike rates? I thought the high threshold argument meant that the opposite should occur (use a high threshold and be highly efficient while emitting few spikes).

We think there is confusion as to multiple possible meanings of ‘efficient’. Cells use 47% of information transmission capacity during gating (vs. 24% ), so that means that they make a more efficient use of the channel capacity of the cell, but that is a different meaning than energy efficiency. In fact, the spike rate more than doubles (factor of ~3.6), so the neural code transmits more information but uses even more energy during gating. So there is a tradeoff. We now spell this out more clearly in the text.

This seems to directly go against the water-filling argument in the Discussion: the analogy there would be that you use the most efficient parts of the channel first (where you get the highest amount of information per spike possible), and only when you need to cram more bits through the channel do you need to use the less efficient parts. So what's going on here?

The water-filling approach to maximizing information indicates that signal power + noise power should be constant for different frequencies (Shannon, 1949) or times (Goldsmith and Varaiya, 1993), except that signal power should be zero if the noise level exceeds that same constant. After a shift, it would be expected that the variance of the input would be greater from the model in Figure 7, and that the SNR of the bipolar cell membrane potential would be higher. This fits with the idea of water-filling, to allocate more power (energy in terms of spikes) to times of higher SNR.

*4) Why don't we see effects of peripheral inhibition, i.e., those putatively responsible for saccadic suppression?*

This is the strong inhibition that follows excitation, now mentioned (subheading “Peripheral input gates central information transmission”).

[Editors' note: further revisions were requested prior to acceptance, as described below.] *[…] A key feature of the model structure is that the bipolar responses are linear and hence that the peripheral input occurs prior to the nonlinearity in the model. The new data added to the paper on bipolar responses do not make a convincing case for linearity of the bipolar responses to natural stimuli. Specifically, while the bipolar responses appear close to linear for 20% contrast noise stimuli (as in new figure), the natural scene stimuli seem very likely to involve much larger changes in input (typical scenes include regions that differ 20-50 fold in intensity). Hence the model involves extrapolation of the bipolar responses well beyond the range where linearity was tested. As noted in the previous round of reviews, photoreceptors are known to adapt strongly to changes in their mean input. It is not clear from the paper what the statistics of the natural inputs are for the model in Figure 7, but it would be surprising if the photoreceptor responses were linear across these stimuli. Published work supports nonlinearities in the photoreceptor responses to natural inputs (Endeman and Kamermans, 2010). Normalizing the mean of the entire image is not adequate to prevent adaptation in the photoreceptors since that process will not change the fact that separate regions differ substantially in intensity, and dynamic adaptation will occur during movement of the image on the retina.The manuscript should be revised either by directly testing the model assumptions, if the authors have those data available, or shifting the model into the Discussion and treating in more like a 'toy' or heuristic model with the assumptions that the model is based on (and caveats to those assumptions) stated clearly.*

As requested, we have moved Figure 7 to the Discussion, and included a paragraph stating the caveats of the model:

“This model does not capture all nonlinearities of the bipolar cell response, including luminance adaptation, a slightly saturating nonlinearity for high contrast stimuli, and weak contrast adaptation. Furthermore, because we did not include luminance adaptation, the model effectively assumes that during the fixation period, adaptation has reached a steady state, and that the global shift is too brief to cause substantial luminance adaptation. The main goal of the model, however, was to gain insight into the dynamics of information transmission under sudden image shifts.”